EMBO
Molecular Medicine

# Fatty acid synthase mediates EGFR palmitoylation in EGFR mutated non-small cell lung cancer

Azhar Ali[1,*,†] (iD), Elena Levantini[2,3,4,†], Jun Ting Teo[1], Julian Goggi[5], John G Clohessy[3], Chan Shuo Wu[1], Leilei Chen[1], Henry Yang[1], Indira Krishnan[3], Olivier Kocher[3], Junyan Zhang[3], Ross A Soo[1,6], Kishore Bhakoo[5], Tan Min Chin[1,7,‡,**] (iD) & Daniel G Tenen[1,2,‡,***] (iD)

## Abstract

Metabolic reprogramming is widely known as a hallmark of cancer cells to allow adaptation of cells to sustain survival signals. In this report, we describe a novel oncogenic signaling pathway exclusively acting in mutated epidermal growth factor receptor (EGFR) non-small cell lung cancer (NSCLC) with acquired tyrosine kinase inhibitor (TKI) resistance. Mutated EGFR mediates TKI resistance through regulation of the fatty acid synthase (FASN), which produces 16-C saturated fatty acid palmitate. Our work shows that the persistent signaling by mutated EGFR in TKI-resistant tumor cells relies on EGFR palmitoylation and can be targeted by Orlistat, an FDA-approved anti-obesity drug. Inhibition of FASN with Orlistat induces EGFR ubiquitination and abrogates EGFR mutant signaling, and reduces tumor growths both in culture systems and *in vivo*. Together, our data provide compelling evidence on the functional interrelationship between mutated EGFR and FASN and that the fatty acid metabolism pathway is a candidate target for acquired TKI-resistant EGFR mutant NSCLC patients.

**Keywords** acquired resistance; EGFR-TKI; FASN; NSCLC; palmitoylation
**Subject Categories** Cancer; Digestive System; Pharmacology & Drug Discovery

## Introduction

Alterations in lipid metabolism are now gaining recognition as a hallmark of cancer cells (Hirsch *et al*, 2010; Patterson *et al*, 2011; Santos & Schulze, 2012). Metabolic pathway reprogramming in

cancer cells has been shown to support cancer cell proliferation and survival (Cairns *et al*, 2011; Hanahan & Weinberg, 2011; Ward & Thompson, 2012). The dependence of tumor cells on deregulated lipid metabolism/biosynthesis indicates that proteins involved in this process are attractive chemotherapeutic targets. In human cells, lipid homeostasis is regulated via a feedback regulatory system by sterol regulatory element-binding proteins (SREBPs) to balance and maintain intracellular levels of fatty acid (FA) and cholesterol (Horton *et al*, 2002). FASN is important in the synthesis of palmitate, precursor of fatty acids, and is found frequently upregulated in many human cancers (Menendez & Lupu, 2007; Ogino *et al*, 2008; Migita *et al*, 2009). FASN is crucial for cancer cell survival, and its overexpression has been associated with poor prognosis, higher risk of cancer recurrence, and drug resistance, in colon and ovarian cancer, breast and renal cell carcinomas (Alo' *et al*, 1996; Gansler *et al*, 1997; Horiguchi *et al*, 2008; Liu *et al*, 2008; Ogino *et al*, 2008). More importantly, enhanced FASN expression and activity has been studied in lung squamous cell carcinomas and reported to occur early in the development and progression of disease (Piyathilake *et al*, 2000).

Despite significant progress in our understanding of the role of lipid metabolism in cancers, there is still poor knowledge on the mechanism/s involved. In addition, the relationship between the roles of epidermal growth factor receptor (EGFR) and lipid signaling remains unclear. With limited effective therapeutic options in non-small cell lung cancer (NSCLC), identification of new targets for therapeutic intervention is sorely needed. In this study, we investigated the mechanisms of resistance to EGFR tyrosine kinase inhibitors (TKIs) in NSCLC. Our data demonstrate a novel EGFR/FASN signaling axis in acquired TKI-resistant NSCLC, which is abrogated by forced EGFR silencing. Persistent EGFR signaling in TKI-resistant EGFR mutated NSCLC cells is shown to be maintained through its

1 Cancer Science Institute of Singapore, National University of Singapore, Singapore City, Singapore
2 Harvard Stem Cell Institute, Harvard Medical School, Boston, MA, USA
3 Beth Israel Deaconess Medical Center, Boston, MA, USA
4 Institute of Biomedical Technologies, National Research Council (CNR), Pisa, Italy
5 Singapore Bioimaging Consortium (A*STAR), Singapore City, Singapore
6 Department of Hematology-Oncology, National University Cancer Institute, National University Health System, Singapore City, Singapore
7 Raffles Cancer Centre, Raffles Hospital, Singapore City, Singapore
 *Corresponding author. Tel: +65 98219600; E-mail: csiazhar@nus.edu.sg
 **Corresponding author. Tel: +65 96357196; E-mail: csictm@nus.edu.sg
 ***Corresponding author. Tel: +65 63112306; E-mail: daniel.tenen@nus.edu.sg
 † These authors contributed equally to this work
 ‡ These authors contributed equally to this work as 14th and 15th authors

palmitoylation. Palmitoylation of EGFR alters its cellular distribution being frequently associated with its nuclear translocation. Further, we observe that EGFR palmitoylation is important to support growth of TKI-resistant EGFR mutated NSCLC cells. Pharmacological inhibition of FASN with Orlistat triggered cell death in TKI-resistant cells in culture and reduced tumor burden in both xenografts and transgenic mouse models. Altogether, these results demonstrate that the fatty acid metabolic pathway is a candidate therapeutic target for the treatment of tyrosine kinase inhibitor resistant EGFR mutant NSCLC.

# Results

## The fatty acid metabolic pathway is upregulated in EGFR mutant NSCLC cells with acquired TKI resistance

To identify resistance-associated pathway signatures, microarray analyses were carried out on parental PC-9 cells (termed PC-9P), which harbor the clinically relevant EGFR delE746-A750 mutation rendering them sensitive to the TKI drug Gefitinib, and resistant PC-9 cells are referred to as PC-9GR. Prior microarray, DNA from PC-9GR was analyzed and tested negative for EGFR T790M mutation and MET amplification (Appendix Fig S1). After characterization of differentially expressed genes, we utilized Gene Set Pathway Enrichment analyses (GSEA) to identify enriched signaling pathways. Interestingly, one of the enriched pathways associated with acquired Gefitinib resistance was linked to fatty acid metabolic pathway. Appendix Table S1 lists the top 10 fatty acid metabolic pathway genes found to be upregulated in PC-9GR when compared to PC-9P cells. These genes were further validated by Q-PCR (Appendix Table S2). Of particular note, FASN, a key regulatory gene for fatty acid synthesis, ranked first in this list. Another highly upregulated gene was sterol regulatory element-binding transcription factor 1 (SREBF1), encoding sterol regulatory element-binding protein 1 (SREBP1), which is the major transcriptional regulator of FASN gene. To determine whether the fatty acid pathway is important for TKI-resistant EGFR mutated NSCLC growth, we evaluated the effect of Orlistat, a FASN inhibitor, on the viability of Gefitinib-resistant EGFRdelL747-A749/A750P HCC4006GR (negative for T790M and MET amplification; Appendix Fig S1), EGFR-delE746-A750 H1650, and EGFRdelE746-A749/T790M H820 cells. Viability assays showed that FASN inhibition through Orlistat could effectively suppress growth of these TKI-resistant cells (Appendix Fig S2). Together, these results suggest that the fatty acid pathway is important, and its upregulation might contribute to acquired TKI-resistant phenotype.

## Acquired TKI-resistant in EGFR del746-750 NSCLC cell growth shows elevated cellular free fatty acid and FASN expression, which is suppressed by silencing EGFR

To investigate whether a link exists between constitutively active EGFR signaling and growth of cells with Gefitinib resistance, both PC-9P and PC-9GR cells were exposed to either Gefitinib or EGFR siRNAs for 72 h. From hereon, the Gefitinib doses used for isogenic PC-9 cells were 50 nM and 1 μM for both PC-9P and PC-9GR cells, respectively. Gefitinib, at 50 nM, is selected as it represents the IC50 value of PC-9P while Gefitinib at 1 μM represents the highest clinically achievable plasma level for the treatment of Gefitinib-resistant

cells (Cohen *et al*, 2004). Since PC-9GR cells were insensitive to Gefitinib, EGFR siRNAs was utilized to inhibit EGFR expression, and the efficacy of EGFR silencing was determined by Western blotting (Appendix Fig S3A). Proliferation and viability assays demonstrated that while Gefitinib failed to suppress growth of PC-9GR cells, EGFR knockdown strongly decreased their survival and growth (Fig 1A and B). To determine the specificity of EGFRi#1 siRNAs, we first generated an EGFRi#1 resistant turboGFP (tGFP)-tagged EGFR cDNA (EGFRdel386-391) through deleting 12 of 21 nucleotides of the EGFR sequence targeted by EGFRi#1 siRNAs by site-directed mutagenesis (Appendix Fig S3B). The specificity of EGFR knockdown by EGFRi#1 was determined by transfecting tGFP-tagged EGFR or tGFP-tagged EGFRdel386-391 into EGFRi#1-treated NL20 cells, followed by measurement of EGFR mRNA using tGFP-specific primers. Quantitative PCR assays showed that tGFP-tagged EGFRdel386-391 was resistant to EGFR knockdown by EGFRi#1 siRNAs, when compared to tGFP-tagged EGFR (Appendix Fig S3C). Furthermore, cell cycle analysis showed that both Gefitinib exposure and EGFR silencing increased the apoptotic sub-G1 population in PC-9P cells while in PC-9GR cells, sub-G1 was elevated only with EGFR silencing (Appendix Fig S3D). We utilized a second EGFR siRNAs (EGFRi#2) to demonstrate similar growth suppression in PC-9GR cells (Appendix Fig S4A). Collectively, these results indicate that EGFR signaling plays a role in the growth and survival of EGFR mutant cells with acquired Gefitinib resistance that can no longer be targeted by Gefitinib.

Given our microarray data demonstrating the upregulation of FA metabolic genes in acquired TKI-resistant EGFR mutant NSCLCs, we examined the relationship between EGFR signaling and fatty acid synthesis. Therefore, cellular free fatty acids (FFAs) were measured in both PC-9P and PC-9GR cells (Fig 1C). In PC-9P cells, cellular FFAs remained unchanged with Gefitinib exposure, EGFR knockdown, or combination of Gefitinib and EGFR knockdown (for 72 h), when compared to vehicle-treated cells. FFAs level was about 45% higher in PC-9GR, as compared to PC-9P cells. In PC-9GR cells, Gefitinib exposure led to a significant elevation of cellular FFAs by about 70%, as compared to vehicle (Fig 1C). Furthermore, EGFR silencing of Gefitinib-treated PC-9GR cells, achieved by two independent siRNAs (Fig 1C; Appendix Fig S4B), demonstrated reduced cellular FFAs to levels similar to those of vehicle-treated cells. To rule out that FFA elevation is not a consequence of Gefitinib off-target effects, we utilized Erlotinib, a different EGFR-TKI, followed by the analysis of cellular FFAs levels in Erlotinib-treated PC-9 cells. Similar to that for Gefitinib, PC-9GR cells exhibited resistance to Erlotinib when compared to PC-9P cells (Appendix Fig S5A). Similar to Gefitinib, cellular FFAs were elevated in Erlotinib-treated PC-9GR cells while FFAs levels were unaffected in Erlotinib-treated PC-9P, when compared to vehicle-treated cells (Appendix Fig S5B). The data suggest that upregulation of cellular FFAs in PC-9GR cells is an EGFR-mediated pro-survival mechanism in response to EGFR-TKI exposure.

To investigate the relationship between EGFR and fatty acid regulation in cells with acquired Gefitinib resistance, we examined SREBP1 and FASN protein expression in PC-9P and PC-9GR cells exposed to Gefitinib for 48 and 72 h. In PC-9P cells, Gefitinib exposure led to significant reduction of EGFR, SREBP1, and FASN expression, after 72 h. However, in PC-9GR cells, EGFR expression remained similar to vehicle after 48 and 72 h of Gefitinib exposure,

**Figure 1. Growth of TKI-resistant EGFR delE746-A750 PC-9 NSCLC cells is associated with elevated cellular free fatty acid and expression of FASN, and is suppressed by EGFR knockdown.**

Gefitinib doses used for these experiments were 50 nM for parental and 1 μM for Gefitinib-resistant PC-9 cells.

A,B   Proliferation ($n = 3$, A) and viability ($n = 3$, B) assays on cells exposed to Gefitinib or EGFR siRNAs for 72 h. Significance in differences in proliferation and viability indexes, in which vehicle acted as control, was determined by $t$-test. Error bars denote SEM.

C   Measurement of cellular free fatty acid (FFA, $n = 3$) after Gefitinib treatment or EGFR knockdown for 72 h. Significance in differences in cellular FFA, normalized against PC-9P vehicle, was determined by $t$-test. Error bars denote SEM.

D   Western blot analysis of cells treated with either vehicle (72 h) or Gefitinib (48 and 72 h).

E   Western blot analysis of cells exposed to either scrambled or EGFR siRNAs (EGFRi#1) for 72 h. ACTB was assessed as housekeeping gene. Vehicle cells acted as controls.

while SREBP1 expression and FASN expression were both upregulated at 48 h and remained elevated at 72 h (Fig 1D). Overall, these data indicate that EGFR positively regulates SREBP1 and FASN expression in acquired TKI-resistant EGFR mutated NSCLC.

Next, given that EGFR was causing upregulation of SREBP1 and FASN, we also evaluated the effect of EGFR knockdown on SREBP1 and FASN expression in PC-9P and PC-9GR cells. EGFR silencing by two different siRNAs was shown to strongly reduce SREBP1 and FASN levels in both PC-9P and PC-9GR cells as compared to untreated and scrambled-treated cells (Fig 1E; Appendix Fig S4C). Moreover, we demonstrated that fatty acid synthesis could play a crucial role in supporting the growth of acquired TKI-resistant EGFR

mutated HCC4006GR cells (Appendix Fig S6). We observed findings similar to these in PC-9GR, in that the viability of Gefitinib-untargetable HCC4006GR was suppressed by EGFR knockdown (Appendix Fig S6A). Cellular FFAs were found higher by about 20% in HCC4006GR, as compared to parental HCC4006P cells (Appendix Fig S6B). Gefitinib exposure induced cellular FFAs elevation by about 40% in HCC4006GR, while FFAs levels remained the same in HCC4006P cells, when compared to vehicle-treated cells (Appendix Fig S6B). Silencing of EGFR in Gefitinib-treated HCC4006GR cells led to reduced FFAs to levels similar to vehicle (Appendix Fig S6B). Western blot analyses showed that SREBP1 and FASN protein expression upon Gefitinib treatment in

HCC4006GR but not in HCC4006P cells (Appendix Fig S6C). Further, HCC4006GR cells were resistant to Erlotinib in cell viability assays (Appendix Fig S7A), and cellular FFAs were significantly elevated with Erlotinib exposure, when compared to HCC4006P cells (Appendix Fig S7B). The result showed that the induced FFAs seen in HCC4006GR cells were a consequence of TKI resistance, therefore eliminating any potential Gefitinib off-target effects. Collectively, these results show that EGFR is involved in the regulation of fatty acid synthesis via SREBP1/FASN and can be inhibited by ablation of EGFR by siRNAs in EGFR mutant NSCLC cells with acquired TKI resistance.

## EGFR-FASN signaling is active in TKI-resistant EGFR T790M/L858R NSCLC cells

In order to ascertain whether the EGFR-FASN regulatory network was active in other Gefitinib-resistant NSCLC cells, carrying either wild-type EGFR or different EGFR mutations, we examined H1975 (EGFR T790M/L858R) and H1703 (EGFR wild-type) cells (Appendix Table S3). Proliferation and viability assays demonstrated that while Gefitinib was weakly effective in Gefitinib-resistant H1975 and H1703 cells, EGFR silencing significantly reduced proliferation and viability of H1975 but less so in H1703 cells (Fig 2A and B).

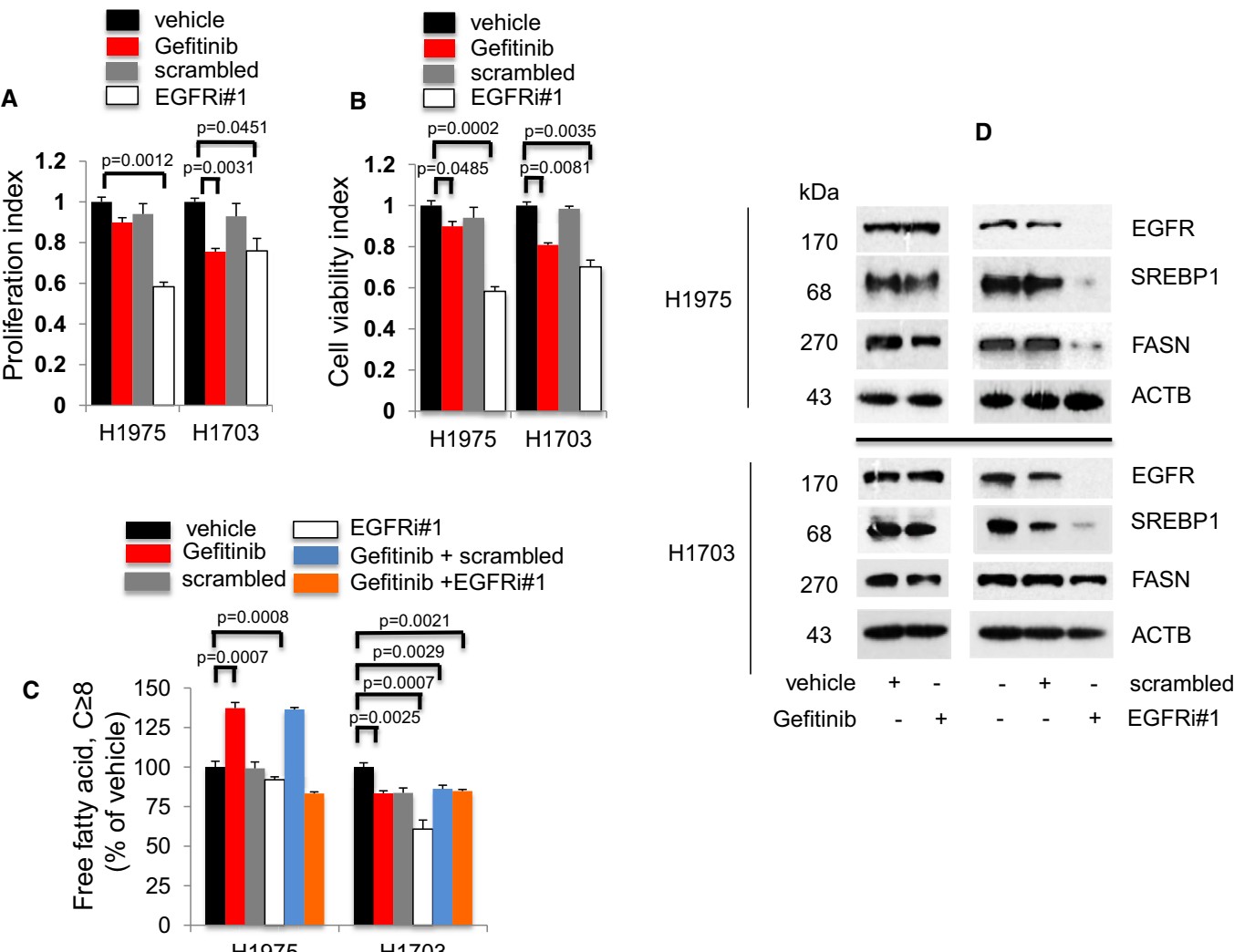

**Figure 2. The EGFR-FASN signaling network is active in TKI-resistant EGFR T790M/L858R NSCLC cells.**

Gefitinib doses used for these experiments were 1 μM, and EGFR siRNAs (EGFRi#1) were at 25 nM.

A, B Proliferation (*n* = 3, A) and cell viability (*n* = 3, B) assays to determine the effects of Gefitinib or EGFR knockdown exposure on EGFR T790M/L858R H1975 and EGFR wild-type H1703 cells after 72 h. Significance in differences in proliferation and viability indexes, in which vehicle acted as control, was determined by *t*-test. Error bars denote SEM.

C     Measurement of cellular free fatty acid (FFA, *n* = 3) after Gefitinib or EGFR knockdown for 72 h. Significance in differences in cellular FFA, in which vehicle acted as control, was determined by *t*-test. Error bars denote SEM.

D     Western blot analysis of H1975 and H1703 cells. Cells were treated with vehicle or Gefitinib for 72 h (left panel). Cells were exposed to either scrambled siRNA or EGFR siRNAs for 72 h (right panel). Vehicle or scrambled-treated cells acted as controls. ACTB was selected as a housekeeping gene control.

Source data are available online for this figure.

These results indicate that the Gefitinib-untargetable EGFR signaling pathway is active in EGFR T790M/L858R NSCLC cells, and important for the survival of EGFR mutated but less so for EGFR wild-type cells.

Therefore, we hypothesized that FFA-downstream pathways might contribute to signals which are unresponsive to Gefitinib. FFAs measurement in Gefitinib-resistant cells demonstrated that Gefitinib exposure induced significantly higher cellular FFAs in Gefitinib-resistant EGFR mutant H1975 cells, which could be abrogated by combined Gefitinib and EGFR knockdown when compared to vehicle (Fig 2C). These molecular changes were not detected in EGFR wild-type H1703 cells (Fig 2C). Additionally, Erlotinib was weakly effective in both H1975 and H1703 cells (Appendix Fig S7C). Moreover, Erlotinib induced cellular FFAs in H1975 cells but not in H1703 cells, thus eliminating FFAs elevation to be a Gefitinib off-target effect (Appendix Fig S7D). Overall, these data demonstrate that the regulation of cellular FFAs by EGFR is restricted to Gefitinib-resistant NSCLC cells carrying EGFR mutations.

Western blot analysis demonstrated the effects of EGFR gene knockdown on SREBP1 and FASN protein expression. In Gefitinib-treated H1975 cells, EGFR, as well as SREBP1 and FASN expression, remained relatively unchanged compared to vehicle, whereas EGFR knockdown resulted in significant loss of SREBP1 and FASN expression, as compared to scrambled-treated cells (Fig 2D upper panels). In H1703 cells (carrying wild-type EGFR), Gefitinib exposure did not affect EGFR, SREBP1, or FASN expression when compared to vehicle, whereas EGFR knockdown led to an almost complete loss of EGFR and SREBP1 expression, leaving FASN expression unaltered when compared to scrambled-treated cells (Fig 2D lower panels). To determine whether the unresponsiveness of SREBP1 and FASN expression in H1703 cells is attributed to the lack of EGFR activity due its wild-type status, we investigated the effects of EGFR hyperactivation by EGF on SREBP1 and FASN expression in these cells. Western blot data showed that SREBP1 and FASN expression remained unaffected after EGFR activation, as demonstrated by the induced EGFR phosphorylation at Y1068 (Appendix Fig S8). Therefore, these results demonstrate that EGFR mutant proteins, not the wild-type form, regulate SREBP1 and FASN expression in Gefitinib-resistant H1975 NSCLC cells.

### Akt signaling is active in acquired TKI-resistant EGFR mutated NSCLC cells and is a critical EGFR downstream target pathway

The Akt pathway is known to be a major downstream effector of EGFR signaling implicated in cell survival. To investigate whether the pro-survival Akt signaling plays a role in acquired Gefitinib resistance, we examined the effects of Gefitinib treatment and EGFR knockdown in PC-9P and PC-9GR, as well as H1975 and H1703 cells by Western blot analysis. Gefitinib treatment resulted in strong reduction of EGFR, Akt, SREBP1, and FASN expression in PC-9P cells (Appendix Fig S9A). However, EGFR, SREBP1, and FASN expression remained relatively unaffected, as well as Akt activity, in Gefitinib-resistant PC-9GR, H1975, and H1703 cells. Knockdown of EGFR led to the abrogation of EGFR, SREBP1, and FASN expression, as well as Akt activity, in PC-9P, PC-9GR, and H1975 cells. Akt pathway activity remained intact in H1703 (wild-type) cells even after Gefitinib exposure or EGFR knockdown. This result suggests that pro-survival Akt is active in Gefitinib-resistant EGFR mutant NSCLC cells and is abrogated by EGFR knockdown.

To validate the importance of Akt, we examined the effect of Akt knockdown on the survival of NSCLC cells. The efficacy of Akt knockdown in PC-9P, PC-9GR, H1975, and H1703 cells was determined by Western blot (Appendix Fig S9B). Results from viability assays demonstrated that Akt silencing significantly reduced cell viability (Appendix Fig S9C), indicating that Akt is important for the growth of EGFR mutant and wild-type TKI-resistant NSCLC cells.

### Overexpression of EGFR mutants prevents growth arrest in EGFR knockdown NSCLC EGFR mutated cells with TKI resistance by re-establishing FASN signaling

To determine whether EGFR mutation status was important in the regulation of FASN for survival of TKI-resistant NSCLC cells, isogenic PC-9 cells as well as H1975 and H1703 cells were selected, as they possess distinct EGFR status and response to Gefitinib exposure. These cells were treated with either EGFR siRNAs alone or in combination with transfection of tGFP-tagged EGFR del746-750, EGFR L858R/T790M, or EGFR wild-type vectors. Cellular proliferation assays showed that overexpression of EGFR mutant vectors prevented growth arrest by EGFR knockdown in TKI-resistant EGFR mutant PC-9GR and H1975 cells (Fig 3A), as compared to EGFR wild-type H1703 cells, in which overexpression failed to prevent EGFR knockdown-mediated growth arrest. Overexpression of EGFR del746-750 in EGFR wild-type H1703 and EGFR wild-type in EGFRdel746-750 PC-9P cells, however, did not increase cell growth (Fig 3A). Similarly, cell viability assays showed increased viability after EGFR mutant overexpression in TKI-resistant PC-9GR (EGFR del746-750) and H1975 (EGFR L858R/T790M), but not in TKI-sensitive PC-9P (EGFR del746-750) or TKI-resistant H1703 (EGFR wild-type) cells (Appendix Fig S10). These results suggest that the EGFR-mediated regulation on cell viability and proliferation, which we have shown to be linked to FASN, is confined to NSCLC cells carrying an EGFR mutation.

In order to confirm that the increased growth by EGFR mutant overexpression in EGFR knockdown cells was indeed linked to cellular FFAs levels, we performed cellular FFAs assays. EGFR mutant overexpression significantly induced FFAs in EGFR knockdown PC-9GR and H1975 cells (both carrying EGFR mutations), whereas EGFR wild-type overexpression did not differ from the scrambled control-treated cells in its ability to induce FFAs in these cells (Fig 3B). Further, neither EGFR mutant nor wild-type overexpression affected FFAs in EGFR knockdown H1703 (EGFR wild-type) or PC-9P (EGFR mutant) cells, as compared to scrambled control. These results highlight a correlation between cellular FFAs and EGFR-mediated FASN signaling in Gefitinib-resistant EGFR mutated NSCLC cells.

The elevated growth rate observed after EGFR mutant overexpression correlated with SREBP1 and FASN expression in EGFR knocked down EGFR mutants PC-9GR and H1975 cells. EGFR mutant overexpression, however, was not able to increase FASN expression in EGFR knockdown PC-9P cells. In EGFR wild-type H1703 cells, FASN levels were similar to scrambled siRNAs after EGFR wild-type overexpression (Fig 3C). These results demonstrated that overexpression of EGFR mutant in EGFR silenced cells correlated with SREBP1 and FASN re-expression in Gefitinib-resistant EGFR mutated cells only. Together, the data suggest that the increased cell growth imparted by EGFR mutant overexpression

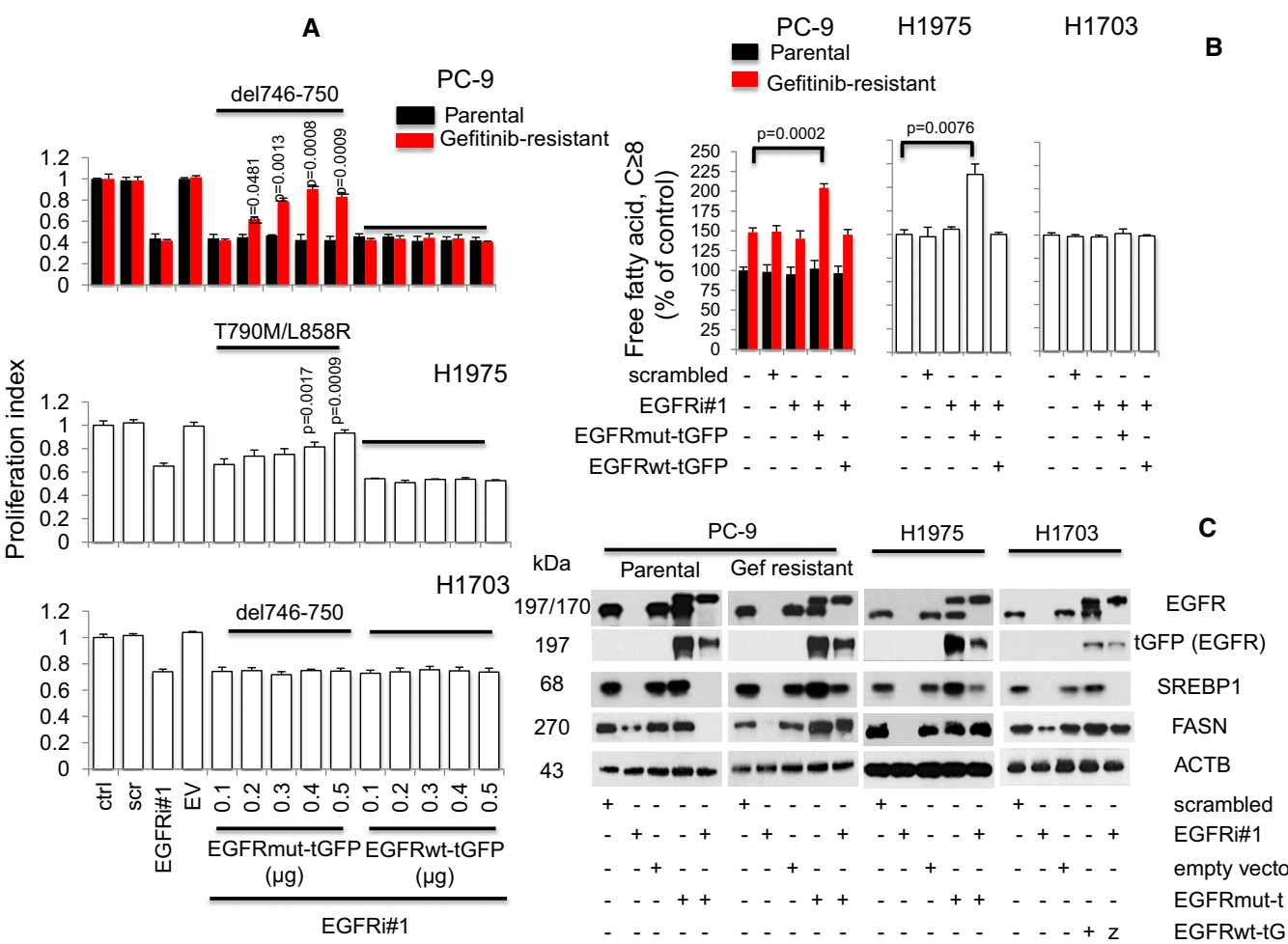

**Figure 3. EGFR mutant overexpression in EGFR knocked down TKI-resistant EGFR mutant NSCLC cells re-establishes FASN and prevents growth arrest.**

A   Proliferation assays (*n* = 3) of tGFP-tagged EGFR (wild type, ΔE746-A750, or L858R/T790M) overexpression in EGFR silenced cells. Cells were allowed to attach overnight in 96-well format prior to transfection with EGFR siRNAs (EGFRi#1, 25 nM) for 12 h. This was followed by transfection with tGFP-tagged EGFR constructs ranging from 0.1 to 0.5 μg in increments of 0.1 μg for additional 60 h (total 72 h). Significance in differences in proliferation index, in which EGFRi#1 cells acted as controls, was determined by *t*-test. Error bars denote SEM.

B   Cells were treated with either scrambled, EGFRi#1, EGFRi#1, and tGFP-tagged EGFR mutant combination, or EGFRi#1 and tGFP-tagged EGFR wild-type combinations for 72 h. Cellular free fatty acid (*n* = 3) was then measured, and significance was determined by *t*-test with untreated cells as control. Error bars denote SEM.

C   Western blot data showing EGFR, exogenous tGFP-tagged EGFR, SREBP1, and FASN expression in NSCLC cells, respectively. Cells were grown in 100-mm dishes overnight prior to EGFR siRNAs (EGFRi#1) transfection for 12 h. This was followed by transfection of 6 μg of respective tGFP-tagged EGFR constructs with further incubation of 60 h (total 72 h). ACTB was selected as housekeeping gene.

Source data are available online for this figure.

correlates with the re-establishment of EGFR-FASN signaling axis, and this phenomenon is exclusive to Gefitinib-resistant EGFR mutant NSCLC cells.

**Palmitoylation of EGFR influences its cellular distribution and is important for TKI-resistant EGFR mutant NSCLC cell growth**

Palmitoylation is the covalent attachment of palmitate to cysteine residues of protein (Linder & Deschenes, 2007; Blaskovic *et al*, 2013; Resh, 2016), which influences the localization, function, and stability of modified proteins (Greaves & Chamberlain, 2007; McCormick *et al*, 2008; Kong *et al*, 2013; Coleman *et al*, 2016). The

addition of saturated 16-carbon palmitate to EGFR has been previously reported (Macdonald-Obermann & Pike, 2009; Bollu *et al*, 2015; Runkle *et al*, 2016). Since we identified an EGFR-mediated regulation of FASN in TKI-resistant EGFR mutant NSCLC cells, we then asked whether EGFR is a target for palmitoylation. To determine this, we performed the *in vitro* acyl-biotin exchange (ABE) assay to detect palmitoylation of EGFR in NSCLC cells. TKI-sensitive cells (PC-9P), together with TKI-resistant (PC-9GR and H1975) NSCLC and NL20 cells, immortalized non-transformed lung cells carrying wild-type EGFR, were treated with Gefitinib to determine whether response to TKIs was linked to EGFR palmitoylation. The efficiency of EGFR immunoprecipitation for their respective cell

lines was assessed before proceeding with the assay (Appendix Fig S11). As shown in Fig 4A, Gefitinib exposure led to a strong upregulation of EGFR palmitoylation in TKI-resistant PC-9GR, and weaker induction in H1975 cells when compared to vehicle. In TKI-sensitive cells, EGFR palmitoylation was greatly reduced by Gefitinib, while EGFR palmitoylation was undetectable in NL20 control cells. These findings indicate a tight correlation between EGFR palmitoylation and TKI resistance in EGFR mutant cells.

Earlier, we have shown that the reversal of growth arrest after EGFR mutant overexpression in EGFR knockdown TKI-resistant EGFR mutants PC-9GR and H1975 cells was accompanied by EGFR re-expression. We then asked whether the re-expressed EGFR was palmitoylated, and we were able to observe that indeed re-expression of mutant EGFR correlated with palmitoylation in both PC9-GR and H1975 NSCLC cells (Fig 4B). To ascertain whether mutated EGFR is preferentially palmitoylated, we performed *in vitro* palmitoylation of tGFP-tagged EGFR under cell-free conditions. *In vitro* translated tGFP-tagged wild-type, EGFR del746-750, and EGFR L858R/T790M proteins were incubated with cellular extracts from PC-9GR, H1975, and H1703 cells followed by ABE assay and Western blot. In Fig 4C, Western blot images showed that IVT EGFR del746-750 and L858R/T790M proteins were preferentially palmitoylated when incubated in PC-9GR and H1975 lysates, while IVT EGFR wild-type proteins remained unpalmitoylated. None of the IVT tGFP-tagged EGFR proteins was palmitoylated after incubation with H1703 lysates. Therefore, palmitoylation appears to be exclusively confined to mutant EGFR suggesting that, in TKI-resistant NSCLC, the structure of mutant EGFR and/or interaction with effector protein/s may have influenced its palmitoylation.

We then asked whether EGFR palmitoylation influences its cellular distribution/localization in TKI-resistant NSCLC cells carrying EGFR mutations. We performed nuclear, cytosolic, and membrane fractionation to compare EGFR distribution between TKI-resistant PC-9GR and H1975, as well as TKI-sensitive PC-9P cells. Cellular fractionations were then subjected to co-immunoprecipitation of EGFR followed by *in vitro* ABE assay to detect EGFR palmitoylation.

From Fig 4D, Western blot images showed that EGFR was predominantly localized in cytosolic and membrane fractions of both TKI-resistant (PC-9GR and H1975) and sensitive (PC-9P) vehicle-treated cells. However, we observed EGFR in the nuclear fractions of vehicle-treated TKI-resistant PC-9GR and H1975 cells, which was not evident in the TKI-sensitive PC-9P cells. Interestingly, Western blot analysis of palmitoylated EGFR revealed that palmitoylated EGFR could be detected only in the membrane fractions of TKI-sensitive PC-9 cells. Palmitoylated EGFR was detected mainly in the nuclear and cytosolic fractions of TKI-resistant PC-9GR and H1975 cells, and much less so in the membrane fractions. Palmitoylation inhibition resulted in disappearance of EGFR in the nuclear fractions of 2-bromopalmitic acid (2-BP)-treated PC-9GR and H1975 cells (Fig 4D). The data suggest that palmitoylation appears to influence EGFR distribution. More specifically, there is a correlation between EGFR palmitoylation and its nuclear translocation in TKI-resistant EGFR mutated NSCLC cells. Next, to determine whether palmitoylation is important for the survival of TKI-resistant EGFR mutated cells, we carried out rescue assay with the supplementation of palmitate after 2-BP treatment in PC-9P, PC-9GR, H1975, and NL20 cells. Results from cell viability assays showed that exogenous palmitate improved the viability of 2-BP exposed PC-9GR and H1975, but not PC-9P and NL20 cells (Fig 4E). This finding suggests that palmitoylation is required specifically for the survival of TKI-resistant EGFR mutated NSCLC cells.

Three cysteine residues—Cys797, Cys1049, and Cys1146, located within the EGFR cytoplasmic tail were identified to be important sites for its palmitoylation (Bollu *et al*, 2015; Runkle *et al*, 2016). To investigate whether there is a functional role of EGFR palmitoylation, we created two separate palmitoylation-deficient tGFP-tagged EGFR mutant constructs—carrying del746-750 and L858R/T90M, respectively, in which cysteine residues 797, 1,049, and 1,146 were mutated to alanine. The palmitoylation-deficient EGFR constructs were then transfected into PC-9GR (del746-750) and H1975 (L858R/T790M) cells, followed by co-immunoprecipitation of EGFR and *in vitro* ABE assay. The loss of palmitoylation expressed by these

---

**Figure 4.  Palmitoylation of EGFR alters its cellular distribution and is crucial for growth of TKI-resistant EGFR mutated NSCLC cells.**

A   Western blot analysis showing EGFR palmitoylation levels in PC-9P, PC-9GR, H1975, and NL20 cells treated with Gefitinib for 72 h. Hydroxylamine (HAM) is a strong reducing agent that cleaves palmitate from cysteine residues and is necessary for biotinylation. The omission of HAM cleavage (HAM-) serves as negative control for ABE assay.

B   Western blot data showing EGFR palmitoylation in cells treated with either scrambled, EGFRi#1, EGFRi#1 plus tGFP-tagged EGFR mutant combination, or EGFRi#1 plus tGFP-tagged EGFR wild-type for 72 h. tGFP-tagged EGFR is about 197 kDa, while EGFR is 170 kDa.

C   Western blot analysis showing palmitoylation of mutated EGFR under cell-free conditions. Cell-free synthesized tGFP-tagged EGFR was generated by *in vitro* translation using vectors containing EGFR wild-type or mutants (del746-750 or L858R/T790M), with no DNA/template as negative control. Fresh extracts of PC-9GR, H1975 and H1703 cells were prepared in hypotonic buffer. For *in vitro* palmitoylation, synthesized tGFP-tagged EGFR was incubated in cell extracts for 1 h at 37°C, followed by ABE assay and Western blotting.

D   Vehicle- and 2-bromopalmitic acid (50 μM of 2-BP)-treated cells were harvested and cellular fractionation was carried out to obtain nuclear, cytosolic, and membrane fractions and a part of these fractions were analyzed by Western blotting to detect EGFR. The remaining fractions were subjected to EGFR pulldown by anti-EGFR antibody and *in vitro* ABE assay. Palmitoylation of EGFR was then determined by Western blot analysis.

E   Cell viability assays (*n* = 3) of cells were first treated with 250 μM of 2-BP for 24 h, followed by the supplementation of 10 μM of palmitate and a further incubation for 48 h. Significance in differences in viability index, in which 2-BP-treated cells acted as controls, was determined by *t*-test. Error bars denote SEM.

F   Growth curves (*n* = 3) illustrating the proliferation index of cells treated with scrambled, EGFR siRNAs (EGFRi#1) alone or in combination with transfection of tGFP-tagged EGFR mutant (del746-750 or L858R/T790M) constructs with or without C797A/C1049A/C1146A mutations for 72 h. Significance in differences in proliferation index, in which EGFRi#1 cells acted as controls, was determined by *t*-test. Error bars denote SEM.

G   Cells were transfected with tGFP-tagged EGFR mutant (del746-750 or L858R/T790M) constructs with or without C797A/C1049A/C1146A mutations and harvested after 72 h. Cellular fractionation was performed to separate nuclear, cytosolic, and membrane fractions. Fractionates were then analyzed by Western blot analysis to detect the distribution of tGFP-tagged EGFR with anti-tGFP antibody.

Source data are available online for this figure.

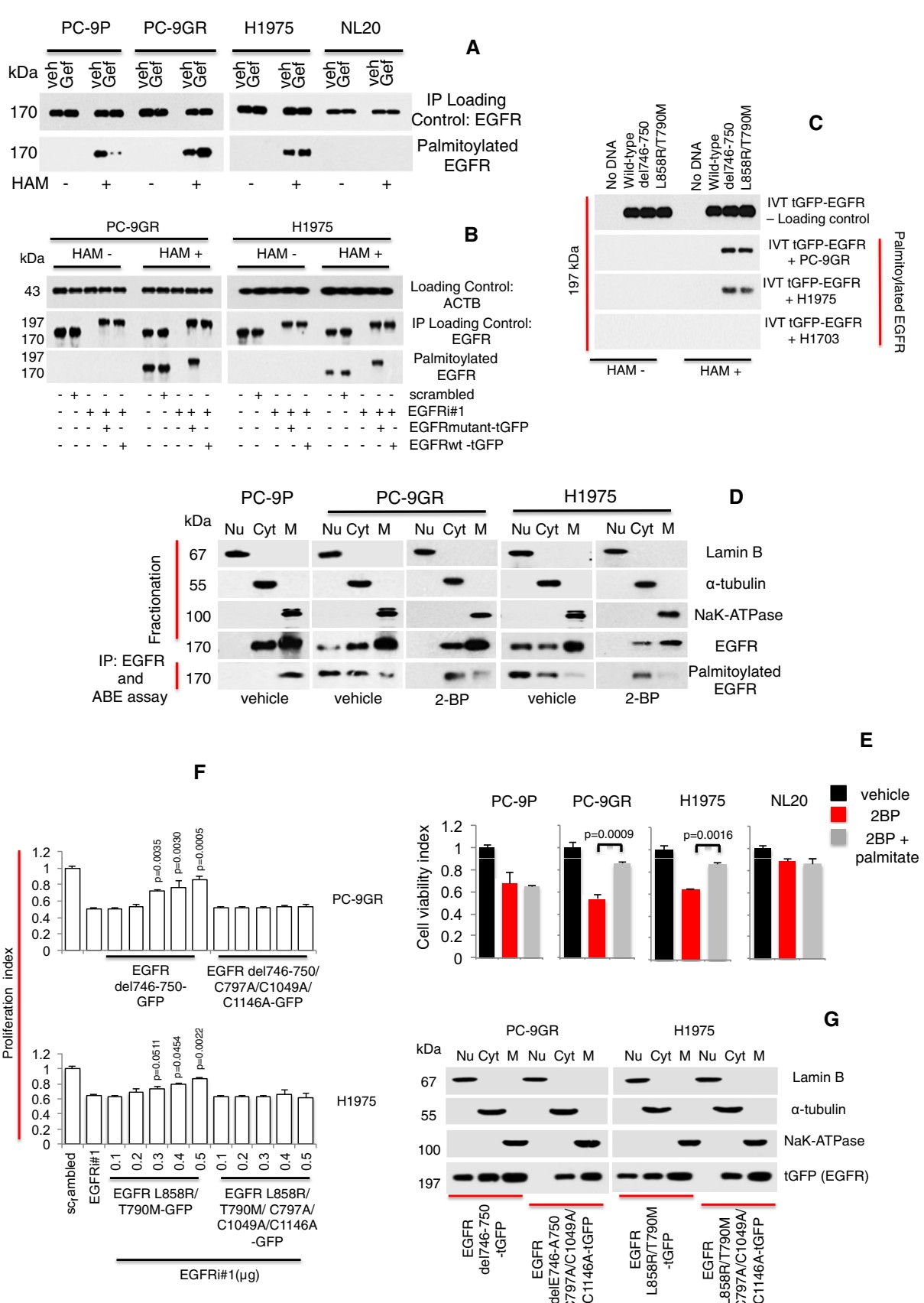

**Figure 4.**

palmitoylation-deficient tGFP-tagged EGFR mutant constructs was first verified by Western blotting (Appendix Fig S12). Results from cell proliferation assays showed that overexpression of palmitoylation-deficient EGFR mutants did not affect the growth decrease in EGFR knocked down PC-9GR and H1975 cells, contrary to cells carrying the solely EGFR mutant constructs (Fig 4F). These data indicate that EGFR palmitoylation is important for the growth and survival of TKI-resistant EGFR mutant PC-9GR and H1975 cells.

Next, we asked whether palmitoylation is necessary for the nuclear translocation of EGFR in TKI-resistant EGFR mutant NSCLC cells. To investigate this, we transfected PC-9GR (del746-750) and H1975 (L858R/T790M) with relevant palmitoylation-deficient tGFP-tagged EGFR mutant constructs, followed by nuclear, cytosolic, and membrane fractionation, and the distribution of tGFP-tagged EGFR mutants was detected by Western blot analysis. Western blot images showed that high EGFR was present in the membrane fractions, with lesser amounts detected in the nuclear and cytosolic fractions of PC-9GR and H1975 cells (Fig 4G). In comparison with tGFP-tagged EGFR, the distribution of palmitoylation-deficient tGFP-tagged EGFR was found predominantly in the membrane fractions of both cells with no changes to EGFR expression in cytosolic fractions while nuclear EGFR was undetectable (Fig 4G). The results suggest that those cysteine residues are important for the nuclear translocation of EGFR in TKI-resistant EGFR mutated NSCLC cells. Together, these data show that the growth of TKI resistance in EGFR mutated NSCLC is linked to EGFR palmitoylation. Further, our data indicate that modification of EGFR by palmitoylation influences its cellular distribution, particularly favoring its nuclear translocation in TKI-resistant EGFR mutated NSCLC cells.

## Pharmacological inhibition of FASN by Orlistat impedes EGFR palmitoylation, enhances its ubiquitination, and suppresses TKI-resistant EGFR mutant NSCLC cell growth

To prove the importance of FASN in Gefitinib-resistant EGFR mutant NSCLC cells, we evaluated the effect of its pharmacological inhibition by using the FASN inhibitor Orlistat. Cellular proliferation and viability in Gefitinib-resistant PC-9GR and H1975 cells were measured by exposing them to increasing drug concentrations, as well as vehicle control (DMSO) for 72 h. The NL20 cell line was included as control. Gefitinib treatment was compared to Orlistat efficacy. TKI-resistant EGFR mutant NSCLC cells exhibited greater sensitivity to Orlistat exposure at 100 μM, indicated by lower growth rates, as compared to NL20 cells (Fig 5A). We then asked whether the lower dependency on FASN is attributed to inactivity of EGFR wild-type in NL20 cells. Viability assays showed that stimulation of EGFR with EGF in NL20 cells did not affect its sensitivity to Orlistat, when compared to EGF alone or vehicle (Appendix Fig S13A). Western blot images demonstrated that EGFR activity (Y1068), SREBP1 and FASN expressions, after Orlistat exposure, remained similar to that of vehicle- plus EGF-treated NL20 cells (Appendix Fig S13B). Together, these results show the efficacy of Orlistat in affecting proliferation and viability of TKI-resistant EGFR mutant NSCLC in cell culture systems.

Next, we examined whether there was a correlation between the inhibitory effects of Orlistat and expression of EGFR, FASN, and apoptotic-related proteins. Western blot data obtained in PC-9GR and H1975 cells demonstrated that Gefitinib exposure failed to

induce Bax and Bak, and concomitantly did not repress survivin, as predicted by their resistance to Gefitinib (Fig 5C). NL20 cells exhibited a similar resistance to Gefitinib. Orlistat exposure, however, led to the upregulation of Bax and Bak, concomitant with loss of EGFR, FASN, and survivin expression in NSCLC cells, as compared to vehicle-treated cells. Conversely, in NL20 cells, Orlistat was able to inhibit FASN expression, however failed to induce Bax and Bak expression, and therefore failed to reduce EGFR and survivin expression. The importance of FASN in Gefitinib-resistant EGFR mutant NSCLC cells was recapitulated with FASN knockdown with siRNAs (Appendix Fig S14A and B). Further, the addition of exogenous palmitate improved cell growth of FASNi-treated PC-9GR and H1975, but not H1703 cells carrying wild-type EGFR (Appendix Fig S15). Together, these data demonstrate that FASN inhibition induces EGFR loss and suppresses the growth of TKI-resistant EGFR mutant NSCLC cells, via inducing apoptosis.

To determine whether FASN was involved in EGFR palmitoylation in TKI-resistant NSCLC cells, we performed *in vitro* ABE assays on Orlistat-treated PC-9GR and H1975 cells. Cells were treated with Orlistat for 24, 48, and 72 h, and their EGFR expression was promptly evaluated by Western blot analysis (Appendix Fig S16). Results showed that Orlistat treatment significantly reduced EGFR palmitoylation (Fig 5D), suggesting a functional relationship between FASN and EGFR palmitoylation in TKI-resistant EGFR mutant NSCLC cells. Further, since palmitoylation has been shown to modulate protein stability by blocking its ubiquitination (Valdez-Taubas & Pelham, 2005; Kong *et al*, 2013), we wondered whether a link between palmitoylation and degradation of EGFR may be observed in TKI-resistant NSCLC cells. Therefore, we sought to determine the effect of Orlistat on EGFR palmitoylation in both PC-9GR and H1975 cells. As shown by Western blot images in Fig 5E, co-immunoprecipitation of EGFR from protein lysates showed that Orlistat induced a dramatic increase in EGFR ubiquitination, whereas EGFR ubiquitination after Gefitinib exposure was similar to vehicle treatment in TKI-resistant cells. These data suggest an inverse relationship between EGFR palmitoylation and ubiquitination.

To evaluate the mechanism through which Orlistat causes cell growth inhibition, we quantified apoptosis by fluorescence activated cell-sorting (FACS) analysis and showed that Orlistat exposure increased the percentages of apoptotic cells in PC-9GR (22.2% ± 1.2% versus 3.4% ± 0.5%, $P = 0.0001$) and H1975 (30% ± 2.4% versus 3.1% ± 0.5%, $P = 0.0001$) as compared to vehicle. The percentages of apoptotic cells were instead similar between Gefitinib- and vehicle-treated cells, consistent with their Gefitinib-resistant phenotype (Fig 5F). Importantly, the percentages of apoptotic cells after Gefitinib or Orlistat exposure were similar to vehicle treatment in NL20 cells. Together, our results demonstrate that Orlistat specifically triggers apoptosis in EGFR mutant Gefitinib-resistant NSCLC cells.

## Orlistat inhibits *in vivo* tumorigenesis in both TKI-resistant NSCLC EGFR mutant xenograft and EGFR-T790M-L858R transgenic models

To determine the effectiveness of Orlistat in inhibiting tumor growth *in vivo*, xenograft models of PC-9GR and H1975 cells were generated in NSG mice. Tumor growth could be detected 5–7 days after

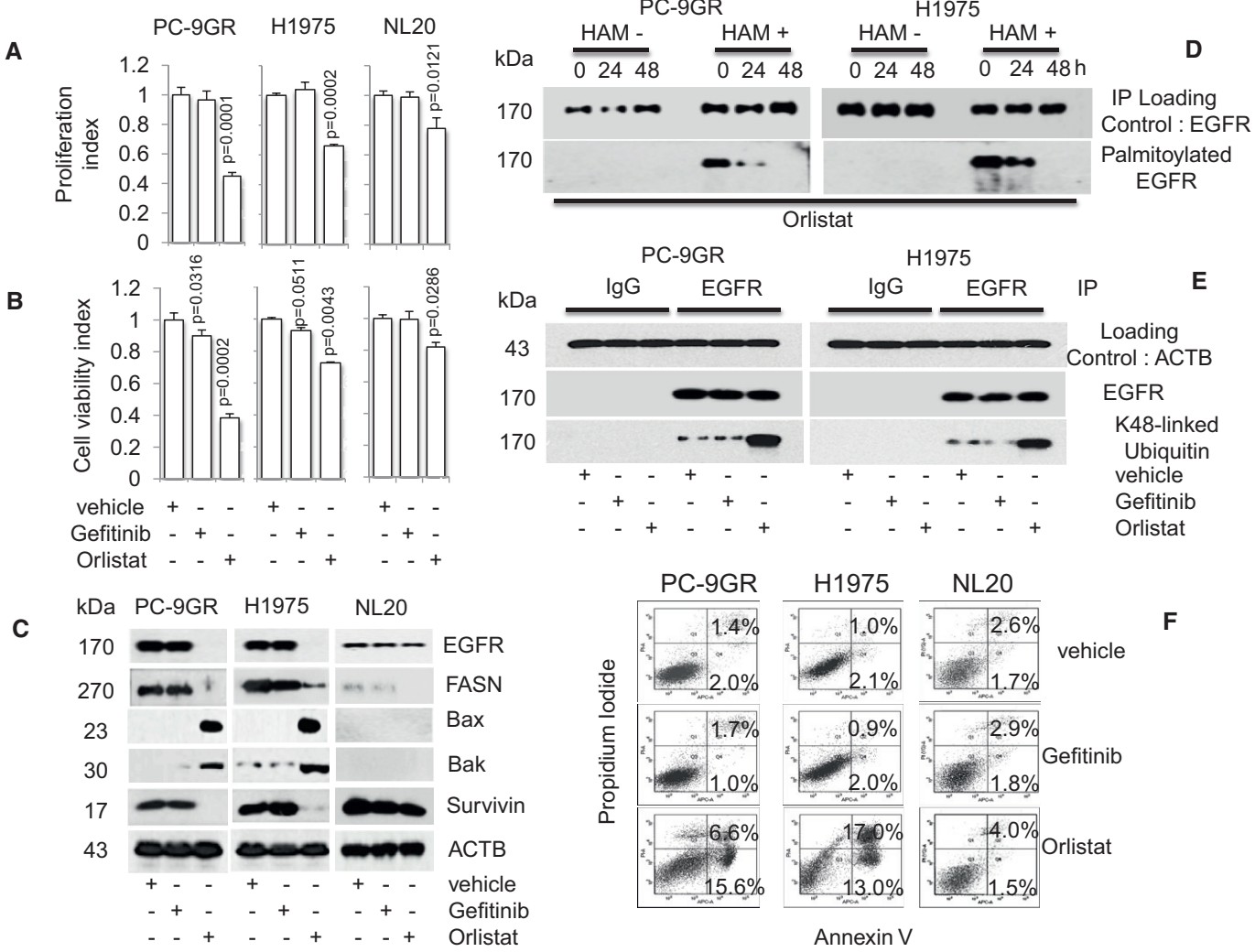

**Figure 5. Orlistat restricts growth in TKI-resistant EGFR mutant NSCLC cells and induces palmitoylation with concomitant enhanced ubiquitination of EGFR.**

A, B  Cell proliferation (*n* = 3, A) and viability (*n* = 3, B) assays on the effect of Orlistat (100 μM) on cells at indicated doses for 72 h (upper and lower panels respectively). Significance in differences in proliferation and viability indexes, in which vehicle acted as control, was determined by *t*-test. Error bars denote SEM.

C      Western blot analysis of EGFR, FASN, Bax, Bak, and survivin protein expression from cells exposed to either Gefitinib or Orlistat for 48 h. ACTB is selected as a housekeeping gene. Vehicle-treated cells acted as controls.

D      Western blot data showing EGFR palmitoylation in Orlistat-treated cells (48 h). Hydroxylamine (HAM), a strong reducing agent that cleaves palmitate from cysteine residues, is necessary for biotinylation. The omission of HAM cleavage (HAM-) serves as negative control for the ABE assay.

E      Western blot images showing ubiquitination status of EGFR in cells after exposure to vehicle, Gefitinib or Orlistat for 48 h.

F      FACS analysis (*n* = 2) for Annexin V/PI staining shows percentages of early (Annexin+) and late (Annexin and PI+) apoptotic cells treated with vehicle, Gefitinib or Orlistat for 48 h.

Source data are available online for this figure.

inoculation, with differences in growth rates occurring after treatment with Orlistat or vehicle (Fig 6A). Tumor volumes of Orlistat-treated mice (*n* = 5 mice) were significantly smaller compared to their vehicle counterparts (*n* = 5 mice). During the course of Orlistat administration, there were no obvious side effects or changes in body weight (Appendix Fig S17). Next, to determine whether Orlistat affects tumor FFAs levels, we measured the FFAs in PC-9GR and H1975 tumor xenografts by ELISA. Results showed that Orlistat significantly reduced FFAs in PC-9GR and H1975 xenograft models as compared to vehicle-treated groups (Fig 6B) Additionally,

Western blotting analysis demonstrated that the decreased growth of Orlistat-treated xenograft tumors correlated with induced Bax and Bak proteins and concomitant loss of EGFR, FASN, and survivin expression, as compared to vehicle-exposed mice (Fig 6C). These data demonstrate that Orlistat is effectively affecting *in vivo* growth of TKI-resistant EGFR mutant NSCLC xenografts.

The *in vivo* efficacy of Orlistat was further examined in the transgenic CCSP-rTTA-EGFR L858R-T790M-driven lung adenocarcinoma mouse model (Politi *et al*, 2006; Li *et al*, 2007). These mice develop lung tumors when continuously exposed to doxycycline, as mutant

EGFR becomes overexpressed in CCSP$^+$ pulmonary epithelial cells. Lung tumor growth was detected and carefully followed by magnetic resonance imaging (MRI). After 5–6 weeks of induction, baseline MRI showed tumor growth in the lungs and at such time point, mice were randomized to vehicle (*n* = 6) or Orlistat (*n* = 7) treatment. MRI images were taken every 2–4 days to capture the effects of drug treatment on tumor size over a 28-day period. Processing and quantification techniques of tumor burden were based on manual segmentation/volume calculation of diffuse lung tumors (Fig 7A) as described previously (Krupnick *et al*, 2012). Changes in lung tumor volumes over the course of treatment were calculated as percentage change in volume over tumor volume at day 1 of treatment, which was set at 100% (Fig 7B). There was no significant difference in tumor volumes between vehicle and Orlistat groups from day 1 to day 23 of treatment. All mice were euthanized at treatment day 28 when tumor burden of vehicle-treated mice became too large. Interestingly, at treatment days 26 and 28, lung

tumors in the Orlistat-treated group were found to be significantly smaller by approximately 30% than those in vehicle-treated (Fig 7B). Tumor FFAs levels analyses showed that the tumor shrinkage in the Orlistat-treated group correlated with significantly lower tumor FFAs when compared to vehicle (Fig 7C). Subsequently, Western blot images showed that FASN expression in tumors from Orlistat-treated transgenic mice showed that tumor growth inhibition correlated with reduced FASN expression when compared to vehicle (Fig 7D). All together, our data show the *in vivo* anti-tumor activity of Orlistat as a single agent against the clinically relevant TKI-resistant EGFR T790M/L858R mutation.

## Discussion

Chemotherapy resistance remains a major problem in limiting the efficacy of conventional and molecular-targeted cancer therapies.

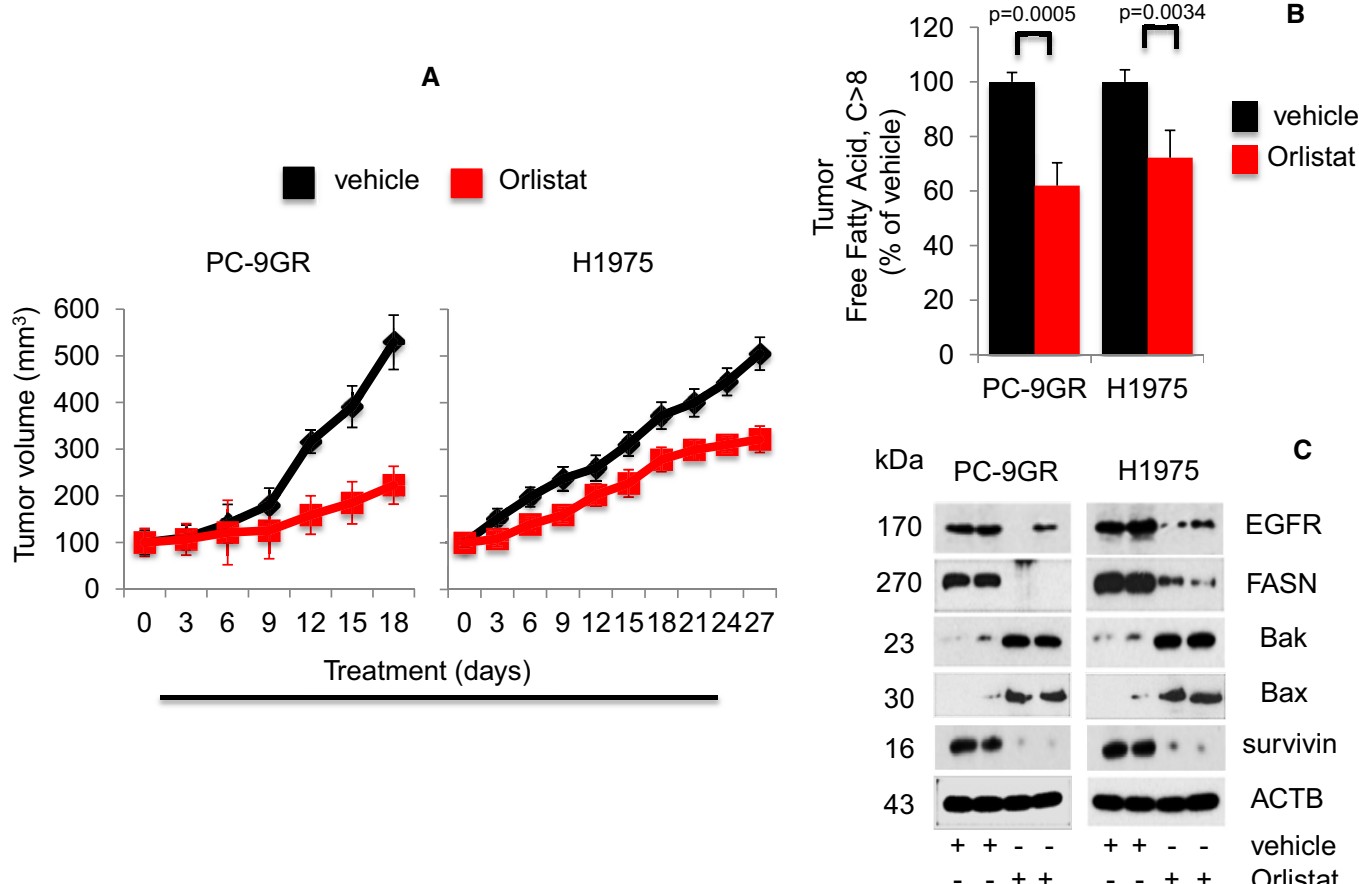

**Figure 6.  Orlistat reduces tumor sizes in PC-9GR and H1975 NSCLC xenograft models.**

A  Growth curve of xenografts from NSG mice treated with either vehicle (*n* = 5) or Orlistat (*n* = 5) at 240 mg/kg body weight (upper panels). Representative images of xenografts from vehicle- and Orlistat-treated mice (lower panels). Significance in differences in tumor sizes, in which vehicle acted as control, was determined by *t*-test. Error bars denote SEM.

B  Measurement of FFAs from tumors derived from vehicle- or Orlistat-treated xenograft mice (*n* = 5 each group). Significance in differences in FFAs, in which vehicle acted as control, was determined by *t*-test. Error bars denote SEM.

C  Western blot analysis of EGFR, FASN, Bax, Bak, and survivin expression in protein extracts from representative xenograft tumors treated with vehicle- and Orlistat-treated groups.

This problem is particularly evident in lung cancer, in which, despite the advances in therapeutic options such as EGFR-TKIs, mortality rate still remains among the highest of cancer-related deaths (Kobayashi *et al*, 2005). Hence, there is an unmet need to identify novel targetable signaling pathways in NSCLC for therapeutic intervention.

To this aim, our initial approach involved chronic adaptation of EGFR del746-750 PC-9 NSCLC cells to increasing doses of Gefitinib over a 6-month duration (Ogino *et al*, 2007; Rho *et al*, 2009). Importantly, DNA analyses from PC-9GR ruled out the link between the development of acquired resistance to Gefitinib of PC-9GR cells and secondary EGFR T790M mutation, which is the most common mechanisms of acquired TKI resistance in NSCLC (Yun *et al*, 2008; Sequist *et al*, 2011). Comparison of global gene expression profiles between PC-9P and PC-9GR cells subsequently uncovered the association of upregulation of fatty acid metabolism pathway genes with

acquired Gefitinib resistance. Most importantly, top-ranked FASN is an attractive molecule for targeted therapy for the following reasons; first, it plays a pivotal role in *de novo* fatty acid synthesis (Guo *et al*, 2009); second, it is found to be overexpressed in various malignancies including those of the lung (Gansler *et al*, 1997; Fiorentino *et al*, 2008; Horiguchi *et al*, 2008; Migita *et al*, 2009); and finally, a clear distinction of fatty acid synthesis exists between adult non-transformed cells and tumors, in that non-transformed cells preferentially use circulating fatty acid (FA) for synthesis of lipids, whereas tumors exhibit elevated *de novo* FA synthesis, irrespective of the extracellular lipid levels (Kuhajda, 2000). Although FASN overexpression has been linked to NSCLC, its association with acquired TKI resistance has not been reported.

A series of experiments unveiled an active regulatory network comprising EGFR and FASN, via SREBP1, that is important for both growth and survival of PC-9GR cells. We have demonstrated that

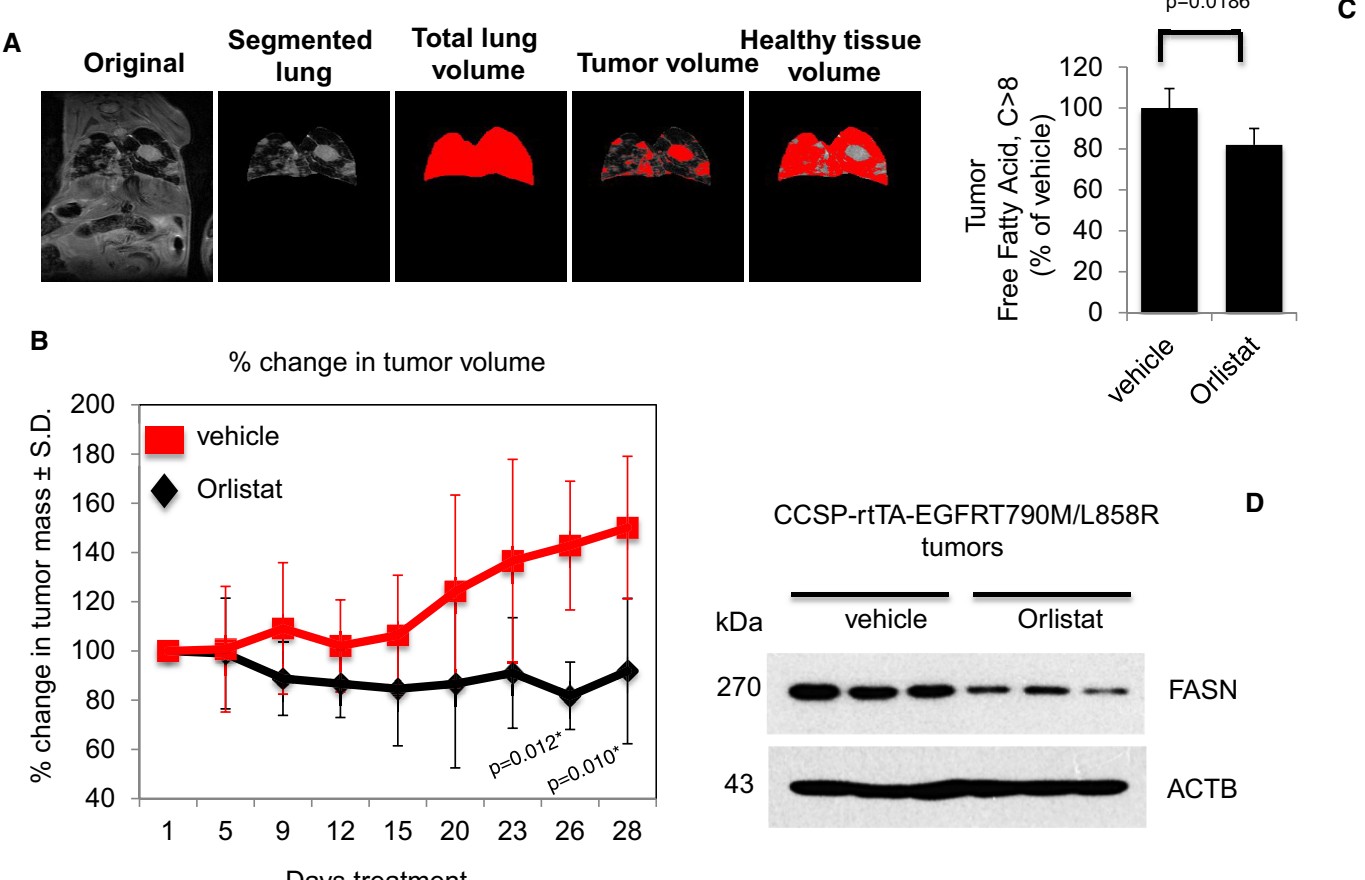

**Figure 7. Orlistat shows anti-tumoral activity in EGFR T790M/L858R transgenic model.**

A   Representative image depicting the determination of tumor burden through manual segmentation. Tumor burden is measured by first defining the lung area, and subsequently determining the average MRI lung image intensity for tumor volume in relation to normal lung volume.

B   Graph shows percentage of change in tumor volume between Orlistat- (*n* = 7) and vehicle-treated (*n* = 6) groups. Significance in differences in tumor volumes, in which vehicle acted as control, was determined by *t*-test. Error bars denote SEM.

C   Measurement of FFAs from tumors of vehicle- or Orlistat-treated transgenic mice (*n* = 5 for vehicle and *n* = 6 for Orlistat groups). Significance in differences in tumor FFAs levels, in which vehicle acted as control, was determined by *t*-test. Error bars denote SEM.

D   Western blot showing FASN expression in protein extracts from representative tumors from vehicle- and Orlistat-treated groups.

Source data are available online for this figure.

despite the presence of Gefitinib, EGFR expression persists and is found to be crucial in the survival of TKI-resistant cells, whereas forced EGFR ablation by siRNAs restricted their growth. Further, the relevance of EGFR-FASN network is uncovered in another acquired resistant EGFR mutated NSCLC cells, HCC4006GR (EGFRdel746-749/A750P), which was generated via the chronic adaptation method. This observation is consistent with an earlier study suggesting EGFR as a valid target in EGFR-TKI-resistant NSCLC cells (Kobayashi *et al*, 2005). Further, in the presence of Gefitinib, persistent EGFR expression correlates with the protein levels of FASN and its activator, SREBP1. Abrogation of FASN and SREBP1 expression by siRNAs-mediated EGFR knockdown demonstrated they are downstream targets of EGFR. FFAs were noticeably elevated in the presence of Gefitinib, which could be lessened to basal levels with EGFR knockdown, suggesting that regulation of cellular fatty acid is EGFR-mediated. The EGFR-FASN link, however, is absent in TKI-sensitive EGFR mutant (PC-9P and HCC4006P) and EGFR wild-type (H1703) NSCLC cells, indicating that the novel molecular link we herein report is confined to Gefitinib-resistant NSCLC cells with EGFR mutant status, and independent of EGFR T790M mutation. An explanation for this observation is the acquisition of additional oncogenic changes in EGFR mutant cells that may converge to tighten the EGFR-FASN axis.

Another important observation of this study was that overexpression of EGFR mutant proteins prevented growth arrest in EGFR silenced cells, by re-establishing FASN expression, and this phenomenon was restricted to Gefitinib-resistant EGFR mutant (PC-9GR and H1975) cells. Furthermore, in these cells, failure to prevent growth arrest through EGFR wild-type overexpression suggests that EGFR mutations are important genotypic determinants in the regulation of FASN. In addition, overexpression of EGFR mutant or wild-type does not prevent growth arrest in EGFR silenced Gefitinib-resistant EGFR wild-type H1703 cells. A plausible explanation for these observations is the involvement of unknown oncogenic signaling pathways activated by EGFR mutants that arose from the adaptation of cells from prolonged exposure to Gefitinib. The Akt signaling pathway is known to be a downstream target of EGFR, and EGFR activating mutations have been reported to deregulate this pathway in cancer (Wong *et al*, 2010). In this study, we have shown that active Akt pathways is a *conditio sine qua non* to allow survival of Gefitinib-resistant EGFR mutant (PC-9GR and H1975) cells. Accordingly, growth arrest is observed after EGFR knockdown causes loss of Akt activity. Furthermore, EGFR signaling through Akt increases the demand of EGFR mutant cells for fatty acid synthesis, resulting in the upregulation of FASN and rapid division of tumor cells (Guo *et al*, 2009; Laplante & Sabatini, 2009; Lamming & Sabatini, 2013).

In this study, we uncover a mechanism of acquired TKI resistance in NSCLC mediated by mutated EGFR in which we demonstrate that EGFR is palmitoylated in PC-9GR and H1975 cells. Protein palmitoylation is the most common acylation of proteins in mammalian cells, and it involves lipid modification of cysteine residues with 16-carbon fatty acid palmitate, which is synthesized by FASN (Kuhajda, 2000; Fiorentino *et al*, 2008; Guo *et al*, 2009; Laplante & Sabatini, 2009; Wong *et al*, 2010; Aicart-Ramos *et al*, 2011; Lamming & Sabatini, 2013). Our data demonstrate that FASN-mediated palmitoylation positively regulates EGFR and is confined to TKI-resistant NSCLC with mutated EGFR. Importantly, our results show that

palmitoylation alters the cellular distribution of EGFR in TKI-resistant EGFR mutated cells. The use of palmitoylation inhibitor or removal of cysteine residues within EGFR cytosolic tail demonstrates that palmitoylation is crucial for the nuclear translocation of EGFR and growth of these cells. Although nuclear EGFR expression has been reported in various malignancies, its expression in highly proliferating normal cells suggests that nuclear EGFR has an important role in normal biology (Brand *et al*, 2011). Further, the identification of nuclear EGFR complexes in various tumor types suggests that downstream target genes of nuclear EGFR, as transcription factor, are distinct and cell type specific (Brand *et al*, 2011).

The functional role of EGFR palmitoylation is presently unclear. Bollu *et al* (2015) reported findings similar to ours that FASN-dependent palmitoylation positively regulates EGFR and supports cancer cell growth whereas Runkle *et al* (2016) showed that palmitoylation suppresses EGFR signaling by "pinning" the C-terminal tail of the receptor to the plasma membrane. Further, Bollu *et al* (2015) showed that palmitoylation is important for plasma membrane localization of EGFR, while Runkle *et al* (2016) demonstrated that palmitoylation inhibition promotes EGFR ubiquitination and prevents its trafficking to the endosomes. In contrast, we demonstrated that palmitoylation is crucial for the nuclear translocation of mutated EGFR. A rationale for the differences seen in the functional impact of palmitoylation on EGFR is that the mutational status of EGFR may influence the functional role of palmitoylation on EGFR, as structural differences exist between wild-type and mutant EGFRs (Yun *et al*, 2007; Lowder *et al*, 2015). It is hence possible that these structural differences contribute to different binding protein partners caused by the underlying mutation within the EGFR tyrosine kinase domain that facilitates its palmitoylation and trafficking.

Pharmacological inhibition of FASN by Orlistat blocks EGFR palmitoylation and affects survival of PC-9GR and H1975 cells. Further, we show that loss of palmitoylation correlates with loss of EGFR protein expression, possibly through ubiquitin-mediated degradation. Orlistat, an anti-obesity drug which targets FASN via its thioesterase domain, has shown to possess anti-proliferative properties and induces apoptosis in tumor cells derived from various malignancies (Pizer *et al*, 1996; Pemble *et al*, 2007; Carvalho *et al*, 2008; Zecchin *et al*, 2011). Accordingly, treatment of PC-9GR and H1975 cells with Orlistat restricted the proliferation and viability of these cells with increased apoptotic cells, activation of apoptosis through upregulation of Bax and Bak, and concomitant loss of EGFR, FASN, and survivin. Additionally, the growth suppression imparted by Orlistat in HCC4006GR, H820, and H1650 cells carrying mutated EGFR strongly suggests that FASN is a valid target in TKI-resistant EGFR mutated NSCLC.

As a proof-of-principle follow-up to our *in vitro* findings, we utilized two murine models to further validate our results in *in vivo* settings. We established NSG xenografts carrying human lung cancer TKI-resistant cells that were subdivided into Orlistat- and vehicle-treated cohorts. According to our *in vitro* data, Orlistat treatment induced significant tumor volume reduction versus vehicle-exposed animals (Agostini *et al*, 2014), in mice carrying EGFR mutant PC-9GR and H1975 xenografts, by 58 and 49%, respectively. Further, reduced tumor size was paralleled by lower FFAs levels, as well as loss of EGFR, FASN, and survivin expression, concomitant to upregulation of Bax and Bak proteins in *in situ* tumors. Orlistat anti-tumor activity

*in vivo* was further validated in transgenic mouse model expressing the clinically relevant EGFR T790M/L858R mutation, in which at termination of treatment, drug-exposed animals presented with approximately 30% decrease in tumor mass (assessed by MRI) as compared to vehicle-treated transgenic mice.

Advances in understanding tumor metabolism in recent years have shown a complex metabolic rewiring necessary for tumor cell adaptations in adverse conditions for survival. Our results are the first to reveal the role of EGFR in controlling FASN signaling in acquired TKI-resistant EGFR mutant NSCLC cells, independent of EGFR T790M mutation. Further, our data demonstrate a functional relationship between EGFR mutant and FASN, in which lipid modification of EGFR by palmitate, the final product of FASN, is important for survival of TKI-resistant cells carrying EGFR mutation, as depicted in Fig 8. However, more research is needed to identify and elucidate how mutant and wild-type EGFR are palmitoylated, and to gain a better understanding on the function/s played by nuclear EGFR in TKI-resistant EGFR mutated NSCLC.

Besides these new avenues of investigation, the important clinical implication of this work is to provide supportive rationale to blocking the fatty acid metabolic pathway in EGFR mutant NSCLC patients, notably in non-responders and/or in those who developed resistance to anti-EGFR therapy. Although the FDA-approved FASN inhibitor, Orlistat, is shown to be effective in reducing tumor cells

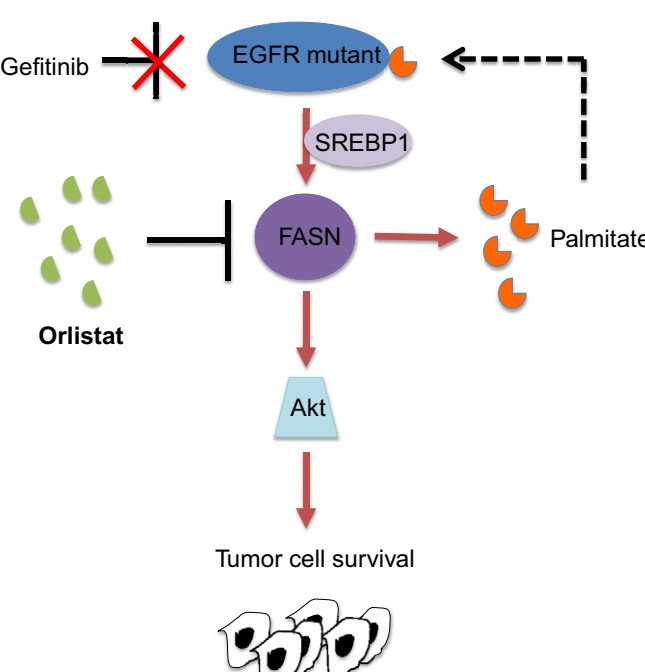

**Figure 8.  Schematic diagram of the proposed model depicting the functional relationship between EGFR and FASN in TKI resistance within EGFR mutant NSCLC cells.**

In TKI-resistant EGFR mutant cells, the Gefitinib-untargetable palmitoylated EGFR positively impinges on FASN, therefore promoting tumor cell growth via the Akt pathway. Palmitoylation is a process where a 16-carbon fatty acid palmitate, synthesized by FASN, is covalently attached to proteins. Pharmacological inhibition of FASN by Orlistat inhibits EGFR palmitoylation and restricts tumor cell survival.

*in vivo,* one main limitation is its extremely low oral bioavailability, as Orlistat exerts its effects primarily in the gastrointestinal tract, where it inactivates pancreatic lipase (McNeely & Benfield, 1998). Improved formulation of Orlistat-like inhibitors of FASN could represent a promising therapeutic modality in the treatment of lung cancer.

# Materials and Methods

### Cell culture

Human lung cancer cells [EGFR mutants (PC-9, HCC4006, H1650, H820, and H1975), EGFR wild-type (H1703) and immortalized non-transformed lung (NL20) were purchased from the American Type Culture Collection (ATCC). These cells tested negative for mycoplasma (Appendix Fig S18; MycoAlert assay from Lonza LT07-218) and were authenticated via DNA fingerprinting (Appendix Fig S19). The mutation status of EGFR for the NSCLC cell lines is summarized in Appendix Table S3. These cells were maintained in RPMI supplemented with 10% FBS (Hyclone) and penicillin/streptomycin (P/S). All cell lines were grown at 37°C in a humidified incubator with 5% $CO_2$. For ubiquitin experiments, MG132 (M7449, Sigma) was added to cells, at a final concentration of 5 μM, 24 h before harvesting to inhibit proteasome activity. Cells were grown to 70–80% confluence, harvested with trypsin, and suspended to the cell density required for each assay.

### Reagents

Short-interfering RNAs targeting EGFR#1 (5′-CAGGAACTGGA TATTCTGAAA-3′), EGFR#2 (5′-AAGTGTGTAACGGAATAGGTA-3′), FASN (5′ – TGGAGCGTATCTGTGAGAA-3′) and AKT (5′-AATCA CACCACCTGACCAAGA-3′) were purchased from Qiagen. Scrambled (non-targeting siRNAs, purchased from Dharmacon (D-001810-01), will be referred to as scrambled in the main text. Anti-EGFR (sc-03, 1:1,000), SREBP1 (sc-8984, 1:1,000), FASN (sc-20140, 1:1,000), β-actin (ACTB, sc-47778, 1:2,000), Lamin B (sc-6217, 1:1,000), and α-tubulin (sc-8035, 1:1,000) antibodies were purchased from Santa Cruz Biotechnology (Santa Cruz, CA). Anti-total Akt (#9272, 1:1,000), phospho-Akt S473 (#9271, 1:1,000), NaK-ATPase (#3010, 1:1,000), K48-linked specific Ubiquitin (#4289, 1:1,000), Bax (#2772, 1:1,000), Bak (#3814, 1:1,000), and survivin (#2808, 1:1,000) antibodies were purchased from Cell Signaling. Anti-tGFP antibody was purchased from Thermo Scientific (PA5-22688, 1:1,000). Anti-phospho EGFR Y1068 was purchased from Abcam (ab5644, 1:2,000). 2-bromopalmitic acid (#238422), palmitate (P9767), and EGF (E9644) were purchased from Sigma. Geftinib (S1025) and Erlotinib (S1023) were purchased from SelleckChem. Xenical capsules (Orlistat) were purchased from Roche and prepared according to Knowles *et al* (2004), for cell culture experiments.

### Microarray

Total RNA prepared from parental and Gefitinib-resistant PC-9 cells was used to probe Illumina HT-12v4 chip. Data analysis was carried out with background correction using Genespring software and was average normalized. Test samples were considered to be

differentially expressed when they crossed the threshold of detection $P$-value $\leq 0.05$ and differential score $P$ value $\leq 0.05$ among all the samples. Significantly differentially expressed genes were annotated with functional assignments to help determine which gene categories were enriched with differentially expressed genes based on the Gene Ontology (GO) database: biological process, molecular function, and cellular component; 1,000 differentially expressed genes were identified at a 1% false discovery rate by SAM analysis (Tusher *et al*, 2001).

## Cell proliferation and viability assays

Cells were plated in 96-well plates at a density of 3,000–5,000 cells/dish in complete medium for 24 h at 37°C in 5% $CO_2$. Cells were then treated with the indicated agents the following day. All samples were assayed in quadruplicates to generate proliferation and viability curves using 5-bromo-2′-deoxyuridine (BrdU assay, #6813, Cell Signaling) and 3-(4,5-dimethylthiazol-2-yl)-5-(3-carboxymethoxyphenyl)-2-(4-sulfophenyl)-2H-tetrazolium) (MTS, G3580, Promega) assays, respectively.

## Gene knockdown

Cells were grown at 50–60% confluence and transfected with 10 or 25 nmol/l of siRNAs using Lipofectamine RNAiMax (#13778, Invitrogen) for 72 h. For negative control, cells were transfected with scrambled siRNAs (non-targeting).

## Mutagenesis and vector transfection

The pCMV6 GFP-tagged EGFR wild-type and mutant (delE746-750 and L858R) constructs were purchased from Origene. The T790M mutation was introduced into the EGFR L858R construct using Quick Change Lightning Multi Site-Directed Mutagenesis kit (#210519, Agilent Technologies) according to the manufacturer's instructions. The mutagenic primer sequences for EGFR L858R are forward 5′-GCATGAGCTGCATGATGAGCTGCACGGTGG-3′ and reverse 5′-CCACCGTGCAGCTCATCATGCAGCTCATGC-3′. For palmitoylation study, the mutagenic primers for are EGFR C797A forward 5′-CTCATGCCCTTCGGCGCCCTCCTGGACTATGT-3′ and reverse 5′-ACATAGTCCAGGAGGGCGCCGAAGGGCATGAG-3′; C1049A forward 5′-CAGCCCATTTCTATCAATGGCAGCCACGGTGGAATTGTTG-3′ and reverse 5′-CAACAATTCCACCGTGGCTGCCATTGATAGAAATGGG CTG-3′ and C1146A forward 5′-GAATGTGCTGTTGACAGCGGTGGGC TGGACAGTG-3′ and reverse 5′-CACTGTCCAGCCCACCGCTGTCA ACAGCACATTC-3′. For EGFRdel391-394, the mutagenic primers are forward 5′-CTCCTCTGGATCCACAGCTGAAAACCGTAAAGGAA-3′ and reverse 5′-TCCTTTACGGTTTTCAGCTGTGGATCCAGAGGAG-3′. All constructs were confirmed by DNA sequencing. Transfection of EGFR-GFP-tagged expression vectors into cells was performed using Fugene HD (E2311, Promega) following the manufacturer's instruction. Briefly, cells (50–60% confluence) were seeded in six-well plates and allowed to attach overnight, and they were incubated with DNA-Lipofectamine mixture comprising 1 μg DNA and 3 μl Fugene 6 reagent diluted in OPTI-MEM (Invitrogen) for 48–72 h. Cells transfected with pCMV6-AC-GFP empty vectors acted as controls. Transfection efficiency was determined visually and by protein expression.

## Quantitative PCR

5 μg of total RNA was reverse-transcribed in a volume of 50 μl using 18-mer oligo-dTs and MMLV reverse transcriptase (Invitrogen). 1 μl of 10-fold diluted reverse transcription reactions was then subjected to PCR in 20 μl reactions with SYBR® Select Mastermix (Life Technologies) and using the primer sequences adapted from PrimerBank (Wang *et al*, 2012) using ABI Prism system (Applied Biosystem). ACTB mRNA levels were measured by quantitative RT–PCR in replicate samples as housekeeping gene for normalization of different mRNA expression and the data are presented as "mRNA Expression". Primer sequences are listed in Appendix Table S4.

## Immunoblot analysis

Post-treatment cultured cells were collected at specific times and solubilized in Cellytic M buffer (C2978, Sigma) with protease and phosphatase (Roche) inhibitor cocktail. Proteins were separated by SDS–PAGE, transferred to PVDF membranes (Thermo), and detected using Western Bright ECL (Advansta, CA).

## Immunoprecipitation

Cells were harvested followed by sonication in lysis buffer [150 mM NaCl, 0.5% NP-40, 1 mM EDTA, 20 mM Tris (pH 7.4), commercial protease and phosphatase inhibitor cocktail] for 10 s. The lysates were cleared by centrifugation and subsequently incubated with relevant or control antibodies for 2.5 h with rotation at 4°C. Protein G beads were then added followed by 1.5 h rotation at 4°C. Immunoprecipitates were then resolved on SDS–PAGE gels.

## Cellular fractionation

Cells were fractionated using the nuclear fractionation kit (#40010, Active Motif) into nuclear, cytosolic, and membrane fractions. Briefly, cells were trypsinized and rinsed twice with PBS containing phosphatase inhibitors. Cell pellets were obtained after centrifugation at 300 $g$ for 5 min and suspended in 1× hypotonic buffer containing phosphatase inhibitors. Cells were incubated on ice for 15 min to allow cells to swell, followed by addition of detergent (supplied by manufacturer), and cells were vortexed vigorously for 10 s. This was followed by immediate centrifugation for 30 s at 14,000 $g$. The supernatant (cytosolic fraction) was transferred to a new eppendorf tube, and pellet (nuclear and membrane fractions) was resuspended in complete lysis buffer. The resuspended pellet was further incubated for 30 min on ice on a rocking platform set at 150 rpm. The suspension was vortexed for 30 s and centrifuged at 14,000 $g$ for 10 min at 4°C. Supernatant (nuclear fraction) was transferred to a new tube, and the pellet (membrane fraction) was resuspended in SDS–PAGE sample buffer.

## *In vitro* acyl-biotin exchange (ABE) assay

The ABE assay is adapted from Wan and colleagues, with modifications (Wan *et al*, 2007). Cells were lysed in lysis buffer (LB pH7.4) containing 50 mM Tris–HCl pH7.4, 150 mM NaCl, 1% NP-40, 1 mM EDTA, and protease inhibitors. For this protocol, all centrifugation steps are carried out at 850 $g$ for 5 min. EGFR was

first immunoprecipitated from 100 μg of protein using 5 μg of anti-EGFR antibody, and complexes of EGFR and anti-EGFR antibody were bound to Protein G Sepharose 4 beads (#17-0618-02, GE Healthcare). 50 mM of N-ethylmaleimide (NEM, E3876, Sigma) in LB pH7.4 was then added to the immunoprecipitated EGFR and incubated 3 h at 4°C with gentle rotation to block free thiols of cysteine residues. After three washes with LB pH7.4, EGFR was treated with and without (mock as control) 1 M of hydroxylamine (HAM, #379921, Sigma) in LB pH7.4 for 2 h at room temperature with gentle rotation. EGFR was then rinsed three times with LB pH6.2 followed by treatment with 5 μM of BMCC-Biotin (#21900, Thermo Fischer Scientific) in LB pH6.2 for 2 h at room temperature with gentle rotation. This was followed by three rinses with LB pH7.4 to remove excess biotin, and EGFR was then eluted with reducing sample buffer. Samples were analyzed by SDS–PAGE gels.

### *In vitro* palmitoylation of cell-free synthesized EGFR

Cell-free tGFP-tagged EGFR vectors, containing wild-type, del746-750 or L858R/T790M, were generated using the 1-Step Human Coupled IVT kit (8882, Thermo Scientific) according to manufacturer's protocol. For fresh cellular extract preparation from PC-9GR, H1975, and H1703, cells were grown on 10-cm dish and are lysed in 500 μl of hypotonic buffer containing 10 mM Tris–HCl pH7.4, 10 mM NaCl, 3 mM $MgCl_2$, 0.1% NP-40. Lysates were passed through 27G needle ten times, and cell debris was removed by centrifugation at 500 $g$ for 5 min at 4°C. Cleared lysates were kept on ice. For *in vitro* palmitoylation, synthesized tGFP-tagged EGFR was incubated with fresh cellular extracts in 2× NT buffer (50 mM Tris–HCl pH7.4 and 300 mM NaCl) for 1 h at 37°C. Half-portion of all reactions were then subjected to ABE assay, and followed by Western blotting. Anti-tGFP antibodies were then used to probe for EGFR.

### Free fatty acid measurement

Total lipids were extracted according to manufacturer guidelines (Free Fatty Acid Quantification Kit, ab65341, Abcam). Briefly, fatty acids were converted to their CoA derivatives and subsequently oxidized with concomitant generation of color. C-8 (octanoate) and longer fatty acids were quantified by colorimetric (spectrophotometry at = 570 nm) with detection limit 2 μM free fatty acid in samples. Relative free fatty acids were quantified based on the cell number ($1 \times 10^6$) and tumor weight (5 mg) for cell lines and tumors, respectively.

### Cell cycle

Cells (both free and attached) were harvested and incubated with propidium iodide solution in PBS at final concentration of 50 ng/ml for 5 min in the dark. Cell cycle analyses were carried out with a FACScan flow cytometer (BD Biosciences). Cell cycle distributions and sub-$G_1$ fractions were calculated using CellQuest software (BD Biosciences, San Jose, CA).

### Flow cytometric analyses

Flow cytometric analyses were carried out on single-cell suspensions and were analyzed by FACScan using CellQuest software (BD Biosciences, San Jose, CA). Apoptosis assay was performed with an Annexin V-FITC Apoptosis Detection Kit #556547 (BD Biosciences, San Jose, CA). Briefly, harvested cells were suspended in 100 μl of the binding buffer followed by additions of 5 μl Annexin V-FITC and 5 μl propidium iodide (PI, 20 mg/ml) and samples were incubated for 15 min at room temperature in dark. Finally, binding buffer (400 μl) was added to each reaction tube and cells were analyzed by flow cytometry.

### Xenograft model

Animal experimental protocols were approved by the NUS Institutional Animal Care and Use Committee (Protocol R14-579). Male NOD scid gamma (NSG) mice were obtained from the Xenograft Cancer Model Facility, Cancer Science Institute Singapore, and maintained in the National University of Singapore Animal Facility throughout this study. PC-9GR and H1975 NSCLC cells were used for injection. There were two groups for each cell line—vehicle- and Orlistat-treated. Mouse numbers were determined using the formula: $n = \log/\log p$, where $n =$ number of animals that need to be tested, 0.05 = the probability of committing a type II error/p = the proportion of the animals in the colony without tumor growth (Dell *et al*, 2002). For our study, we use = 0.05, as we wish to have a 95% chance of detecting the event of tumor formation and we expect the incidence of tumor formation to be 50% in our study. Hence, $P = 0.50$. $n = \log/\log$ p = log 0.05/log 0.50 = 4.33. Mice were randomized, in which even numbered mice were allocated to vehicle group whereas odd numbered mice were allocated to Orlistat group ($n = 5$ for each group). Experimenters were blinded to the treatment groups to avoid bias. Cells were suspended at $3 \times 10^6$ in 200 μl of 1 part PBS: 1 part Matrigel solution (#356234, Corning, MA) and implanted into the right flank of 6- to 8-week-old NSG mice using a syringe with a 27 gauge disposable needle (BD Biosciences). Mice were previously anesthetized with isoflurane by inhalation at 3–4% (Baxter, IL). When tumors reached approximately 4 mm in diameter, mice were treated with either Orlistat (240 mg/kg) or vehicle by intraperitoneal injections for five consecutive days per week. For mice injection, Orlistat was prepared according to Kridel *et al* (2004), with minor modifications. Tumors were measured with caliper every 3 days and their volumes calculated using the formula volume $0.5 \times$ length $\times$ width$^2$. Mice were then sacrificed by $CO_2$ inhalation when tumor reached a diameter of 15 mm in vehicle group or presented with ulceration at tumor site. Tumors were harvested and snap frozen in liquid nitrogen and stored at −80°C.

### CCSP-rtTA-EGFRT790M/L858R transgenic model

All animals were housed in specific pathogen-free housing with abundant food and water under guidelines approved by the BIDMC Institutional Animal Care and Use Committee and Animal Research Facility. *CCSP-rtTA* tetracycline-dependent activator mice (Tichelaar *et al*, 2000) and tet-regulatable *EGFR*$^{L858R + T790M}$ mice (Li *et al*, 2007) have been previously described, were a kind gift from Dr Kwok-Kin Wong, and were maintained on a C57BL/6 genetic background. Calculations to determine mouse numbers are similar to the formula used for *Xenograft model*. However, we expect the incidence of tumor formation to be 40% in the transgenic model. Hence, $P = 0.60$. $n = \log/\log$ p = log 0.05/log 0.60 = 5.86. To induce EGFR expression, doxycycline was administered by feeding

**The paper explained**

**Problem**

Despite the advances in therapeutic options such as targeted therapy using EGFR-TKIs, the mortality rate for lung cancer still remains among the highest of cancer-related deaths and this is largely attributed to the development of acquired resistance to EGFR-TKIs.

**Results**

In this report, we uncovered a novel oncogenic signaling pathway exclusively acting in mutated epidermal growth factor receptor (EGFR) non-small cell lung cancer (NSCLC) with acquired tyrosine kinase inhibitor (TKI) resistance. We show that mutated EGFR mediates TKI resistance through regulation of the fatty acid synthase (FASN), which produces 16-C saturated fatty acid palmitate. Our work demonstrate that the persistent signaling by mutated EGFR in TKI-resistant tumor cells relies on palmitoylation of EGFR and can be targeted by Orlistat, an FDA-approved anti-obesity drug. Inhibition of FASN with Orlistat induces EGFR ubiquitination and abrogates EGFR mutant signaling, and reduces tumor growths both in culture systems and *in vivo*.

**Impact**

With limited effective therapeutic options in TKI-resistant NSCLC, these pre-clinical data offer compelling evidence for the fatty acid metabolism pathway as a candidate target for therapeutic intervention in tyrosine kinase inhibitors resistant EGFR mutant NSCLC.

both male and female mice with doxycycline-impregnated food pellets (625 ppm; Harlan-Teklad). Eight- to ten-week-old mice were fed with doxycycline to induce lung tumors. Mice were then imaged by magnetic resonance (MRI) to detect baseline tumors and subsequently recruited into treatment groups. Utilizing randomization process similar to that for *Xenograft model*, six (two males and four females) and seven (three males and four females) mice were allocated to vehicle and Orlistat treatment groups, respectively. Experimenters were blinded to the treatment groups to avoid bias. The preparation and administration of Orlistat was as described earlier in *Xenograft model*. MRI images of mouse lungs were captured with a Bruker Biospec 94/20 9.4 Tesla scanner, and the primary imaging sequence used was RARE (Rapid Acquisition with Refocused Echoes), with TR/TE = 1,200 ms/17.5 ms.

**Statistical analysis**

Results from cell proliferation and viability assays, Q-PCR and FFAs quantification were from at least two independent experiments and each performed in quadruplicates, are shown as means ± SEM. All statistical analyses, including *in vivo* studies, were performed using *t*-test and $P < 0.05$ was considered statistically significant.

**Data availability**

The Microarray Gene Expression data have been deposited to the Gene Expression Omnibus (GEO) repository (GEO Accession Number: GSE83666).

**Expanded View** for this article is available online.

## Acknowledgements

This work is funded by National University Hospital System NR14IBN003OM; Cancer Science Institute of Singapore R713-001-018-271 and R713-000-216-720; the NCIS Yong Siew Yoon Research Grant through donations from the Yong Loo Lin Trust; the Singapore Ministry of Health's National Medical Research Council under its Singapore Translational Research (STaR) Investigator Award; National Research Foundation Singapore, and the Singapore Ministry of Education under its Research Centres of Excellence Initiative.

## Author contributions

AA, TMC, EL, and DGT designed the experiments, analyzed the data, and wrote the manuscript. AA and JTT carried out the *in vitro* analyses. AA, LC, and JTT carried out the xenograft *in vivo* analyses. AA, CSW, and HY carried out gene profiling. AA, EL, JG, JGC, IK, OK, JZ, RAS, and KB carried out the transgenic *in vivo* analyses.

## Conflict of interest

The authors declare that they have no conflict of interest.

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
