## [Review Process File · EMBO Molecular Medicine]

Fatty Acid Synthase mediates EGFR palmitoylation in EGFR mutated Non Small Cell Lung Cancer

Azhar Ali, Elena Levantini, Teo Jun Ting, Julian Goggi, John G. Clohessy, Wu Chan Shuo, Polly Chen Leilei, Henry Yang, Indira Krishnan, Olivier Kocher, Zhang Junyan, Ross Soo, Kishore Bhakoo, Chin Tan Min, Daniel G. Tenen

Review timeline:

Submission date:	24 July 2017
Editorial Decision:	06 September 2017
Revision received:	12 November 2017
Editorial Decision:	13 December 2017
Revision received:	05 January 2018
Accepted:	11 January 2018

Editors: Roberto Buccione and Céline Carret

Transaction Report:

1st Editorial Decision

06 September 2017

Thank you for the submission of your manuscript to EMBO Molecular Medicine.

I sincerely apologise for the very unusual delay in getting back to you on your manuscript. In fact, we experienced significant difficulties in securing expert and willing reviewers, and then obtaining their evaluations in a timely fashion, mostly due to the overlapping holiday season. Furthermore, I failed to retrieve a third evaluation for this manuscript. To avoid further unproductive delays, I am making a decision based on the two available evaluations.

As you will see, while reviewer 2 provides a thoughtful and detailed critique, the evaluation by #reviewer 1 is positive but not very informative. After internal discussion, it was agreed that reviewer 2 raises many important points that would need to be addressed including 1) the concern that the observed effects of Gefitinib might be due to off-target activity, 2) the need for further evidence to demonstrate that FASN up-regulation in an EGFR resistance cell line is a causal effect, 3) the lack of important controls. Reviewer 2 also notes the need for further experimentation to demonstrate the translational potential of Orlistat re-purposing for NSCLC. We agree that the latter issue needs to be clarified to bring the manuscript to the level expected for an EMBO Molecular medicine article. I wish to stress our impression that notwithstanding the significant elements of concern, reviewer 2 is in essence quite appreciative of the study.

In conclusion, while publication of the paper cannot be considered at this stage, we are willing to consider a substantially revised manuscript, addressing the reviewer's concerns including with further experimentation where required.

***** Reviewer's comments *****

Referee #1 (Remarks for Author):

Ali et al have investigated the TKI-resistance in mutant EGFR NSCLC demonstrating that this is mediated by FASN palmitoylation. Interestingly this reaction can be efficiently targeted in cell culture and in xenografts.

This is an important topic in view of the few treatments available for NSCLC patients. The experiments are well designed, the results clearly written and the figures well understandable.

The AA provide evidence for a functional interrelationship between mutated EGFR and FASN and show that the fatty acid pathway is a candidate target for acquired TKI resistance in EGFR mutant NSCLC patients. This is an important finding that deserves public knowledge at a high level. Moreover, they prove that since the TKI resistant tumor cells rely on EGFR palmitoylation, they can be targeted by Orlistat, an FDA approved anti-obesity drug.

Referee #2 (Remarks for Author):

The authors study the role of palmitoylation in the development of resistance to EGFR TKI inhibitor targeted therapy and the regulation of this palmitoylation by SRBP1 and FASN through EGFR signaling resistance mechanism. They find that, in lung cancer, mutated EGFR mediates TKI resistance through regulation of FASN. The authors also report that TKI resistance can be circumvented by treatment with the FASNi orlistat in cell lines and in transgenic mouse models. The topic is of great interest for several reasons including a totally new pathway to EGFR TKI inhibitor resistance, and also a unique dependency on SRBP1 and FASN that appears to be specific for EGFR mutated lung cancers.

Comments to the Authors;

The paper is reviewed in the context of the potential importance for clinical translation and for elucidation of a new fatty acid synthesis pathway involved in EGFR targeted therapy. The experiments are technically well done and clearly presented. However, there are several issues the authors need to deal with.

1. Several findings are not novel and the primary papers are not discussed. As the author note, it has already been reported that wild type EGFR is palmitoylated. However, the functional role of EGFR palmitoylation is controversial. For instance, Runkle et al, *Mol Cell* (2016) showed that palmitoylation negatively regulates EGFR; palmitoylated EGFR is ubiquitinated and fails to traffic to endosomes. By contrast, Bollu et al. *Oncotarget* (2015), reported that EGFR is activated by FASN-dependent palmitoylation. To clarify these controversial findings, it would be useful if the authors discussed their results in the context of previous literature.
2. The preponderance of the data provided by the authors indicate that EGFR signaling activates fatty acid synthesis (as already documented in the literature), however the authors have not provided sufficient evidence to support their assertion that palmitoylation is required specifically for the signaling of TKI resistant EGFR.
3. Their conclusions are almost entirely based on the study of one EGFR mutant TKI resistant NSCLC line. Because of the importance for clinical translation it is very important to determine if their therapeutic results with orlistat are evident in studies of other NSCLC lines including EGFR mutant, EGFR mutant TKI resistant, and EGFR wildtype NSCLCs. This is particularly true for the xenograft studies. Orlistat may be a potential drug that can be "repurposed" for NSCLC treatment, but the mechanisms may go beyond those focused on in this report and thus could work in a variety of NSCLCs, and/or not work in some EGFR mutant NSCLCs. While they do not have to provide tests on large numbers of xenografts at least some others should be included. Following along this line of thought, for clinical translation, the question naturally arises what if orlistat was added in during the primary xenograft treatment of EGFR mutant NSCLC? Would this prevent the development of resistance? Such data, whichever way it came out, would greatly add to the clinically translational value of this paper.
3. The authors also do not provide sufficient evidence that orlistat reaches its target in their in vivo experiments (see below).

There are also a series of specific points that authors should also deal with that are summarized (not in order of priority):

1. Line 122. The authors should provide the primary data that "DNA from PC-9GR cells was negative from T790M mutations or MET amplification" or cite the appropriate reference;
2. Line 133. The observation that FASN is upregulated in a single EGFR resistant cell line may be a fortuitous event. It would be of interest to know the sensitivity of a larger panel of NSCLC cell lines to FASNi or silencing.
3. Line 140: PC-9 should be named PC-9GR
4. Line 151: given the importance of the knockdown experiments, the phenotype of the RNAi experiment should be "rescued" with an RNAi resistant cDNA (to rule out off-target effects)
5. Line 161: the authors show that gefitinib, but not EGFR silencing, increases FFA in EGFR resistant cells. One interpretation could be that this reflects a gefitinib off target effect rather than by EGFR inhibition. The authors need to discuss. In the same manner it would be useful to show with another EGFR TKI, erlotinib, that their findings hold.
6. Given that all of the experiments relied on an EGFR mutant NSCLC derivative, PC-9Gr, it is possible the effects described could be an idiosyncratic phenomenon that occurs only in PC-9GR. Thus, it would be appropriate for the authors to test other NSCLC EGFR mutant cell lines that acquired gefitinib resistance. In addition, in reviewing the Methods section, there is no statement that the cell lines were DNA fingerprinted or mycoplasma tested. While I think it is highly unlikely that such events occurred, they need to be definitely accounted for. In fact, one event that could explain a lot of their findings, would be mycoplasma contamination of some of the EGFR resistant derivatives.
7. Line 174: The sentence: overall these data show a positive correlation between gefitinib resistance and elevated SREBP1 and FASN" is misleading. A better wording of the sentence would state: "the data indicate that EGFR positively regulates...". This line is also redundant because this concept is also restated at line 181.
8. Line 200 (fig. 2C): again gefitinib increases FFA, but RNAi EGFR knockdown does not cause the same effect. This finding raises the question that FFA elevation is a gefitinib off target effect
9. Line 212. The fact that wild type EGFR does not overtly upregulate SREBP and FASN could be due to the fact that wild type EGFR is not constitutively activated. What is the effect if wild-type EGFR is stimulated with EGF?
10. Line 254; could it be that the lack of growth promoting effects of wild type EGFR is merely due to the fact that it is not constitutively active? In other words, couldn't it be that wild type EGFR stimulates FASN, but at a lower level than constitutively active EGFR? Again EGF treatment may clarify this point.
11. Line 280. EGFR Palmitoylation was already reported to occur. The authors should discuss their findings in relation to the literature.
12. Line 302. The conclusion is based on correlative data. While this would require additional experimentation, structural studies or studies with recombinant proteins in cell free conditions are needed to prove the assertion that mutant EGFR is preferentially palmitoylated.
13. Line 313. The authors should discuss their findings in the context of the literature.
14. Line 365: The conclusion regarding the dependency on FASN is based on the effect of orlistat in NL20 cells, a correlative finding. What happens if NL20 cells are treated with EGF?
15. Line 377 and 395. An important experiment to try (no matter what the result) would be to try to rescue the phenotype of orlistat and/or FASN silencing by administration of exogenous palmitate.
16. Line 408: Given the clinical translational importance of this key experiment, it would be of great value to the paper if the authors could demonstrate that orlistat inhibits FASN in vivo either by performing enzymatic assays or by measuring directly fatty acids in tumors.
17. Line 408 and 452. In my opinion the data presented are insufficient to establish circulating FASN as a biomarker (for instance where is FASN coming from? Exosomes? Why circulating FASN is down-regulated by orlistat treatment?). In my opinion this part is not necessary in the paper.

1st Revision - authors' response

12 November 2017

Referee #1 (Remarks for Author):

Ali et al have investigated the TKI-resistance in mutant EGFR NSCLC demonstrating that this is mediated by FASN palmitoylation. Interestingly this reaction can be efficiently targeted in cell culture and in xenografts.

This is an important topic in view of the few treatments available for NSCLC patients. The experiments are well designed, the results clearly written and the figures well understandable.

The AA provide evidence for a functional interrelationship between mutated EGFR and FASN and show that the fatty acid pathway is a candidate target for acquired TKI resistance in EGFR mutant NSCLC patients. This is an important finding that deserves public knowledge at a high level. Moreover, they prove that since the TKI resistant tumor cells rely on EGFR palmitoylation, they can be targeted by Orlistat, an FDA approved anti-obesity drug.

Referee #2 (Remarks for Author):

The authors study the role of palmitoylation in the development of resistance to EGFR TKI inhibitor targeted therapy and the regulation of this palmitoylation by SRBP1 and FASN through EGFR signaling resistance mechanism. They find that, in lung cancer, mutated EGFR mediates TKI resistance through regulation of FASN. The authors also report that TKI resistance can be circumvented by treatment with the FASNi orlistat in cell lines and in transgenic mouse models. The topic is of great interest for several reasons including a totally new pathway to EGFR TKI inhibitor resistance, and also a unique dependency on SRBP1 and FASN that appears to be specific for EGFR mutated lung cancers.

We would like to thank both reviewers for their comments and recommendations.

Comments to the Authors;

The paper is reviewed in the context of the potential importance for clinical translation and for elucidation of a new fatty acid synthesis pathway involved in EGFR targeted therapy. The experiments are technically well done and clearly presented. However, there are several issues the authors need to deal with.

1. Several findings are not novel and the primary papers are not discussed. As the author note, it has already been reported that wild type EGFR is palmitoylated. However, the functional role of EGFR palmitoylation is controversial. For instance, Runkle et al, Mol Cell (2016) showed that palmitoylation negatively regulates EGFR; palmitoylated EGFR is ubiquitinated and fails to traffic to endosomes. By contrast, Bollu et al. Oncotarget (2015), reported that EGFR is activated by FASN-dependent palmitoylation. To clarify these controversial findings, it would be useful if the authors discussed their results in the context of previous literature.

*We have inserted the requested information in the current version of the manuscript, from **lines 582 to 596** of the discussion, hypothesizing reasons for the differences observed between our study and that of others on the effects of EGFR palmitoylation.*

2. The preponderance of the data provided by the authors indicate that EGFR signaling activates fatty acid synthesis (as already documented in the literature), however the authors have not provided sufficient evidence to support their assertion that palmitoylation is required specifically for the signaling of TKI resistant EGFR.

*Following the Reviewer's suggestion, we performed cell rescue assays, with the supplementation of palmitate after 2BP treatment in PC-9P, PC-9GR, NL20, and H1975 cells to support the evidence that palmitoylation is important for survival of TKI resistant cells (please see **lines 359 to 365; and Fig 4E**).*

3. Their conclusions are almost entirely based on the study of one EGFR mutant TKI resistant NSCLC line. Because of the importance for clinical translation it is very important to determine if their therapeutic results with orlistat are evident in studies of other NSCLC lines including EGFR mutant, EGFR mutant TKI resistant, and EGFR wildtype NSCLCs. This is particularly true for the xenograft studies. Orlistat may be a potential drug that can be "repurposed" for NSCLC treatment, but the mechanisms may go beyond those focused on in this report and thus could work in a variety of NSCLCs, and/or not work in some EGFR mutant NSCLCs. While they do not have to provide tests on large numbers of xenografts at least some others should be included. Following along this

line of thought, for clinical translation, the question naturally arises what if orlistat was added in during the primary xenograft treatment of EGFR mutant NSCLC? Would this prevent the development of resistance? Such data, whichever way it came out, would greatly add to the clinically translational value of this paper.

We have included data obtained from another EGFR mutated NSCLC cell line with acquired TKI resistance, HCC4006GR (EGFRdel746-749/A750P), in which EGFR-FASN signaling is active, whereas the parental line HCC4006P has no such active molecular pathway (Lines 180 to 195; and Fig S6). We treated HCC4006GR, together with the TKI resistant EGFR mutated cell lines H1650 (EGFR del746-750) and H820 (EGFRdel746-750/T790M), with Orlistat to demonstrate its efficacy in TKI resistant EGFR mutated NSCLC (Fig S2). With regards to the question if Orlistat can delay Gefitinib resistance, we thank the Reviewer for this important question that we will address as part of a follow-up study, as it requires a considerable amount of time to obtain this data.

3. The authors also do not provide sufficient evidence that orlistat reaches its target in their in vivo experiments (see below).

We carried out FFA measurement on xenograft tumors (Fig 6B) and transgenic models (Fig 7C), and examined FASN protein expression in transgenic tumors (Fig 7D), demonstrating that Orlistat reaches tumors and affects cellular fatty acid and FASN levels.

There are also a series of specific points that authors should also deal with that are summarized (not in order of priority):

1. Line 122. The authors should provide the primary data that "DNA from PC-9GR cells was negative from T790M mutations or MET amplification" or cite the appropriate reference;

We carried out DNA analyses on EGFR T790M mutation and MET amplification on PC-9GR and HCC4006GR cells and we included these data, as requested in Appendix Fig S1.

2. Line 133. The observation that FASN is upregulated in a single EGFR resistant cell line may be a fortuitous event. It would be of interest to know the sensitivity of a larger panel of NSCLC cell lines to FASNi or silencing.

We tested the sensitivity of TKI resistant EGFR mutated HCC4006GR, H1650, and H820 cells to Orlistat and demonstrated that FASN plays a role in the survival of these resistant cells (Lines 109 to 115; Fig S2).

3. Line 140: PC-9 should be named PC-9GR

We apologize for not making it clear, and that PC-9 is the correct label as isogenic PC-9 cells refers to both parental and resistant cells.

4. Line 151: given the importance of the knockdown experiments, the phenotype of the RNAi experiment should be "rescued" with an RNAi resistant cDNA (to rule out off-target effects)

To address this, we created EGFRi#1, a RNAi resistant EGFR cDNA clone to determine the specificity of EGFRi#1 siRNAs, demonstrating that the siRNAs are specific against their intended target (Lines 131 to 139; Fig S3C).

5. Line 161: the authors show that gefitinib, but not EGFR silencing, increases FFA in EGFR resistant cells. One interpretation could be that this reflects a gefitinib off target effect rather than by EGFR inhibition. The authors need to discuss. In the same manner it would be useful to show with another EGFR TKI, erlotinib, that their findings hold.

To address this issue, we examined the sensitivity of PC-9P, PC-9GR, HCC4006P, and HCC4006GR cells to Erlotinib, and compared cellular FFAs levels between vehicle- and Erlotinib-treated cells. Our data show that the elevated FFAs seen is not a consequence of Gefitinib off-target effects but an observation seen in the resistant cells against TKI inhibitors (Lines 157 to 161; Fig S5A and S5B; and Lines 191 to 195; Fig S7A and S7B).

6. Given that all of the experiments relied on EGFR mutant NSCLC derivative, PC-9Gr, it is possible the effects described could be an idiosyncratic phenomenon that occurs only in PC-9GR. Thus, it would be appropriate for the authors to test other NSCLC EGFR mutant cell lines that acquired gefitinib resistance. In addition, in reviewing the Methods section, there is no statement that the cell lines were DNA fingerprinted or mycoplasma tested. While I think it is highly unlikely that such events occurred, they need to be definitely accounted for. In fact, one event that could explain a lot of their findings, would be mycoplasma contamination of some of the EGFR resistant derivatives.

To address Reviewer's comment, TKI resistant EGFR mutated HCC4006GR, H820, and H1650 cells were included in this study. Acquired resistant HCC4006GR exhibited the EGFR-FASN signaling axis, similar to that in PC-9GR cells. All three cell lines were found resistant to both gefitinib and erlotinib, whereas their growth could be suppressed by FASN inhibition. We tested all cell lines used in this study for mycoplasma (Fig S18) and DNA fingerprinted (Supplementary File 2).

7. Line 174: The sentence: overall these data show a positive correlation between gefitinib resistance and elevated SREBP1 and FASN" is misleading. A better wording of the sentence would state:" the data indicate that EGFR positively regulates...". This line is also redundant because this concept is also restated at line 181.

We have amended the text, as advised (Lines 173 to 174) by rephrasing it into – “.....these data indicate that EGFR positively regulates SREBP1 and FASN expression in acquired TKI resistant EGFR mutated NSCLC.....”

8. Line 200 (fig. 2C): again gefitinib increases FFA, but RNAi EGFR knockdown does not cause the same effect. This finding raises the question that FFA elevation is a gefitinib off target effect.

We tested H1975 and H1703 cells with Erlotinib, another TKI inhibitor, and compared FFAs between vehicle- and Erlotinib-treated cells and demonstrated that the elevated FFAs is not likely due to gefitinib off-target effects (Lines 217 to 219, Fig 7D).

9. Line 212. The fact that wild type EGFR does not overtly upregulates SREBP and FASN could be due to the fact that wild type EGFR is not constitutively activated. What is the effect if wild-type EGFR is stimulated with EGF?

We investigated the effects of EGFR activation by EGF on SREBP1 and FASN expression in H1703 cells (Lines 231 to 238; Fig S8) and demonstrated that EGFR mutant proteins, and not wild-type, regulate SREBP1 and FASN. We thank the Reviewer for this observation that allowed us to strengthen our observation through these carefully designed controls.

10. Line 254; could it be that the lack of growth promoting effects of wild type EGFR is merely due to the fact that it is not constitutively active? In other words, couldn't it be that wild type EGFR stimulates FASN, but at a lower level than constitutively active EGFR? Again EGF treatment may clarify this point.

Please find our response in our point #9 where we demonstrated that EGFR mutant proteins, and not wild-type, regulate SREBP1 and FASN expression.

11. Line 280. EGFR Palmitoylation was already reported to occur. The authors should discuss their findings in relation to the literature.

We did acknowledge reports regarding EGFR palmitoylation (lines 308-310 and 582 to 596). Please also see our response to author query 13.

12. Line 302. The conclusion is based on correlative data. While this would require additional experimentation, structural studies or studies with recombinant proteins in cell free conditions are needed to prove the assertion that mutant EGFR is preferentially palmitoylated.

*We performed cell-free palmitoylation assays to determine if mutant EGFR is preferentially palmitoylated, and our data demonstrated that EGFR mutants are preferentially palmitoylated over wild-type proteins (please see **Lines 329 to 337; and Fig 4C**).*

13. Line 313. The authors should discuss their findings in the context of the literature.

*We followed the Reviewer's concern and discussed these findings in the Discussion section (**lines 582 to 596**).*

14. Line 365: The conclusion regarding the dependency on FASN is based on the effect of orlistat in NL20 cells, a correlative finding. What happens if NL20 cells are treated with EGF?

*To determine if the lack of dependency on FASN is attributed to lack of constitutive EGFR activity in wild-type NL20 cells, cells were stimulated with EGF prior Orlistat treatment, and our data indicate that stimulation of EGFR activity does not affect NL20's lack of sensitivity to Orlistat (please see **Lines 409 to 416; Fig S13A and S13B**).*

15. Line 377 and 395. An important experiment to try (no matter what the result) would be to try to rescue the phenotype of orlistat and/or FASN silencing by administration of exogenous palmitate.

*We carried out rescue assay by supplementation of palmitate to FASN knockdown PC-9GR, H1975, and H1703 cells, and our data demonstrated that exogenous palmitate improved the growth of FASNi-treated cells, thus reinforcing our initial observations (please see **Lines 428 to 432; Fig S15**).*

16. Line 408: Given the clinical translational importance of this key experiment, it would be of great value to the paper if the authors could demonstrate that orlistat inhibits FASN in vivo either by performing enzymatic assays or by measuring directly fatty acids in tumors.

*We measured tumor FFAs from xenograft (**Fig 6B**) and transgenic mice (**Fig 7C**), and examined FASN protein expression in tumors from transgenic mice (**Fig 7D**), demonstrating that Orlistat inhibits FASN and affects FFAs levels in tumors.*

17. Line 408 and 452. In my opinion the data presented are insufficient to establish circulating FASN as a biomarker (for instance where is FASN coming from? Exosomes? Why circulating FASN is down-regulated by Orlistat treatment?). In my opinion this part is not necessary in the paper.

Following the Reviewer's suggestions, we decided to remove our previous statement on circulating FASN as biomarkers.

2nd Editorial Decision

13 December 2017

Thank you for the submission of your revised manuscript to EMBO Molecular Medicine. We have now received the enclosed report from the referee who was asked to re-assess it. As you will see the reviewer is now globally supportive and I am pleased to inform you that we will be able to accept your manuscript pending a few final editorial amendments.

***** Reviewer's comments *****

Referee #1 (Comments on Novelty/Model System for Author):

This is the second review. The improvements made in the manuscript make it very novel, highly technical and high on medical impact. The system is adequate this is at a pre-clinical level.

Referee #1 (Remarks for Author):

As requested by the Editor I have scrutinised the manuscript not only following my personal

criticisms but also, and in depth, those raised by reviewer 2.

The authors have done an immense amount of experimentation answering all the points raised. I feel extremely satisfied for their answers and for their their experiments and for their modifications of the manuscript.

The only point that the AA do not address is whether "Orlistat can delay Gefitinib resistance". I agree with the AA that the amount of time required required for this experiment makes it unsuitable at the level of a manuscript review.

In addition, I believe that this is not a question that should have been asked, as the answer would belong to a totally different project.

Corresponding Author Name: Dr Daniel Tenen
Journal Submitted to: EMBO Molecular Medicine
Manuscript Number: EMM-2017-08313